# Effects of residue substitutions on the cellular abundance of proteins

**Thea K Schulze, Kresten Lindorff-Larsen\***

The Linderstøm-Lang Centre for Protein Science, Department of Biology, University of Copenhagen, Copenhagen, Denmark

## eLife Assessment

This **valuable** study presents a thorough analysis of protein abundance changes caused by amino acid substitutions, using structural context to improve predictive accuracy. By deriving substitution response matrices based on solvent accessibility, the authors demonstrate that simple structural features can predict abundance effects with accuracy comparable to complex methods such as free energy calculations. The strength of the evidence is **convincing**, supported by robust experimental design and comprehensive analyses.

**\*For correspondence:**
lindorff@bio.ku.dk

**Abstract** Multiplexed assays of variant effects (MAVEs) make it possible to measure the functional impact of all possible single amino acid residue substitutions in a protein in a single experiment. Combination of variant effect data from several such experiments provides the opportunity to conduct large-scale analyses of variant effect scores measured across proteins, but can be complicated by variations in the phenotypes that are probed across experiments. Thus, using variant effect datasets obtained with similar MAVE techniques can help reveal general rules governing the effects of amino acid variation for a single molecular phenotype. In this work, we accordingly combined data from six individual variant abundance by massively parallel sequencing (VAMP-seq) experiments and analysed a total of 31,614 variant effect scores reporting solely on the impact of single amino acid residue substitutions on the cellular abundance of proteins. Using our combined variant effect dataset, we derived and analysed a collection of amino acid substitution matrices describing the average impact on cellular abundance of all residue substitution types in different structural environments. We found that the substitution matrices predict the cellular abundance of protein variants with surprisingly high accuracy when given structural information only in the form of whether a residue is buried or exposed. We thus propose our substitution matrix-based predictions as strong baselines for future abundance model development.

## Introduction

The 20 commonly occurring amino acids have vastly different physicochemical properties, which allow them to play distinct roles for protein structure formation and function. Some amino acid residues might be particularly important for the thermodynamic and cellular stability of a certain protein and others crucial for intrinsic activity or function, with the different amino acid residue types contributing to each of these properties in unique ways (*Ribeiro et al., 2020*; *Dunham and Beltrao, 2021*; *Tsuboyama et al., 2023*; *Cagiada et al., 2024*). For certain aspects of protein functionality, the role played by each type of amino acid to maintain this functionality is well understood. For example, the propensity of each of the 20 amino acids to form or break secondary structural elements has been studied extensively. The typical way to assess the role played by different amino acids in proteins is by using mutagenesis studies. The experimental methods available for performing such studies have

developed immensely, and the field has thus progressed from investigating single or few residue substitutions at a time to conducting alanine scans to now using saturation mutagenesis methods, including MAVEs, that allow the impact of thousands of residue substitutions on protein structure and function to be studied in a single experiment (*Fowler and Fields, 2014*).

The wealth of mutagenesis data collected in, for example, MAVE experiments can be used to analyse substitution effects in individual proteins comprehensively. The data also provides the opportunity to conduct large-scale studies to uncover general rules regarding the properties and functions of amino acid residue types across proteins, if data reporting on substitution effects in multiple proteins is combined. Some MAVEs assess the effects of residue substitutions on several proteins in a single experiment (*Rocklin et al., 2017*; *Tsuboyama et al., 2023*; *Beltran et al., 2024*), thus naturally allowing the latter type of analysis. However, most studies investigate substitution effects in a single protein at a time, and metaanalysis thus requires that data from different individual experiments are combined (*Gray et al., 2017*; *Dunham and Beltrao, 2021*; *Høie et al., 2022*). Such metaanalyses have proven very useful and have, among other things, revealed 100 amino acid subtypes based on substitution effect profiles (*Dunham and Beltrao, 2021*) and provided insight into which amino acid types that best discriminate protein-ligand binding interface from non-interface residues across proteins (*Gray et al., 2017*).

While combining substitution effect scores from several MAVEs allows for creation of large datasets, such an approach must take into consideration that the term MAVE covers a range of experimental methods that seek to investigate different aspects of protein function and use a variety of techniques for this purpose (*Weile and Roth, 2018*; *Tabet et al., 2022*; *Geck et al., 2022*). MAVE methods share a common foundation, namely setting up a system linking protein sequence or genotype to a phenotypic readout that can be selected for on large scale and traced back to genotype via sequencing (*Fowler and Fields, 2014*; *Fowler et al., 2014*). However, the molecular phenotype being studied and the selection mechanism used change from assay to assay. Some studies might take interest in variant effects on enzymatic activity or ligand binding affinity, while others might be concerned with the impact of substitutions on cellular fitness, to give a few examples. Ultimately, these assays are thus expected to report on overlapping but non-identical variant effects.

MAVEs are additionally performed in various experimental backgrounds, for instance in different organisms and at different temperatures, which influence measured variant effect scores. It has been demonstrated that changing a single parameter related to the experimental background, for example expression level (*Jiang et al., 2013*; *Cisneros et al., 2023*) or the chemical (*Mavor et al., 2016*; *Mavor et al., 2018*; *Weile et al., 2021*) or genetic (*Thompson et al., 2020*; *Nguyen et al., 2024*) background, can result in substantial differences to the MAVE readout. In an aggregated dataset consisting of a variety of MAVE scores, phenotype- or background-specific effects will thus likely be present and add noise to the analysis performed with the dataset. A final obstacle for MAVE score combination is that the effects of substitutions are quantified in different ways across experiments (*Peterman and Levine, 2016*; *Rubin et al., 2017*), meaning that score combination typically requires implementation of one (*Gray et al., 2017*; *Munro and Singh, 2021*; *Høie et al., 2022*) or several (*Dunham and Beltrao, 2021*) score transformation schemes to make scores from individual datasets compatible (or at least be on the same scale).

As the number of published MAVE datasets has grown (*Esposito et al., 2019*; *Rubin et al., 2021*), it has become increasingly feasible to perform metaanalyses of combined datasets that were obtained with similar methodologies and report on the same molecular phenotypes. On this basis, it is possible not only to decrease potential background noise but also to discover general rules governing residue substitution effects for that specific phenotype. Thus, inspired by several previous MAVE data-based, large-scale analyses of substitution effects (*Gray et al., 2017*; *Dunham and Beltrao, 2021*) and the recent availability of homogeneous MAVE data, we here analysed a collection of six MAVE datasets that all report on the impact of single residue substitutions on protein cellular abundance. Our analysis is thus focused on a single molecular phenotype, cellular abundance, which is known to be crucial for protein function and at the same time easily impacted by substitution of single residues, with widespread consequences for human disease (*Matreyek et al., 2018*; *Cagiada et al., 2024*).

Variant cellular abundance has been studied with different MAVE setups, including through protein fragment complementation assays that rely on monitoring cell growth (*Faure et al., 2022*) and by using fluorescent reporters to estimate cellular concentration (*Matreyek et al., 2018*). The

substitution effects analysed here were all measured using the same type of MAVE experiment, specifically VAMP-seq, and the analysed effects were thus all observed in highly similar experimental backgrounds and were quantified with the same selection mechanism using almost identical methods (*Matreyek et al., 2018*; *Matreyek et al., 2021*; *Amorosi et al., 2021*; *Suiter et al., 2020*; *Grønbæk-Thygesen et al., 2024*; *Clausen et al., 2024*). The VAMP-seq method estimates the steady-state abundance of all single residue substitution variants of a protein through expression of green fluorescent protein-tagged protein variants in cultured human cells. Each cell expresses a single protein variant, and the cellular fluorescence intensity thus reports on the cellular abundance of the expressed variant. Abundance scores for thousands of variants are obtained in high-throughput by separating cells into bins of discrete fluorescence intensity intervals using fluorescence-activated cell sorting (FACS) followed by sequencing of each bin. Expression is controlled for, and changes in abundance are therefore mostly a product of changed degradation behaviour due to the substitution of residues (*Matreyek et al., 2018*).

The six VAMP-seq datasets have all been analysed extensively and separately in previous work (*Matreyek et al., 2018*; *Matreyek et al., 2021*; *Amorosi et al., 2021*; *Suiter et al., 2020*; *Cagiada et al., 2021*; *Grønbæk-Thygesen et al., 2024*; *Clausen et al., 2024*). Here, we combined the six datasets and thereby obtained a total of 31,614 abundance variant effects scores, allowing us to perform a large-scale analysis of the impact of single residue substitutions on cellular abundance across proteins and thus contribute to our understanding of how different amino acid residues help maintain cellular protein levels. In our analysis, we specifically focused on analysing and predicting the impact of residue substitutions on abundance using only simple considerations of the structural environment of residues. Simple structural descriptors, such as local packing density and side-chain accessible surface area, have previously been shown to capture some substitution effects on protein folding stability, for example for cavity-creating substitutions (*Serrano et al., 1992*; *Chakravarty and Varadarajan, 1999*; *Cota et al., 2000*). The impact of substitutions on protein stability can to some extent also be predicted when a simple structural descriptor like residue relative solvent accessibility is used in combination with information about the physicochemical properties of wild-type and variant residues (*Caldararu et al., 2021*). Since substitutions that affect the free energy of folding of a protein tend to also have an impact on cellular abundance (*Nielsen et al., 2017*; *Scheller et al., 2019*; *Abildgaard et al., 2019*; *Matreyek et al., 2018*; *Suiter et al., 2020*; *Cagiada et al., 2021*; *Høie et al., 2022*; *Gerasimavicius et al., 2023*; *Grønbæk-Thygesen et al., 2024*; *Clausen et al., 2024*), similar structural descriptors might also be useful for analysis of cellular stability substitution effects. Our goal in this work is ultimately thus not to develop the most accurate possible variant abundance predictor, but rather to show how much variation in cellular abundance can be understood by simple rules that apply across different proteins.

For our structure-based analysis of the combined VAMP-seq dataset, we constructed a set of amino acid substitution matrices that for different structural environments contain the average abundance scores for all possible residue substitution types. Others have in earlier work similarly calculated substitution matrices using experimental mutagenesis data and studied the ability of the matrices to predict the impact of mutations (*Yampolsky and Stoltzfus, 2005*; *Munro and Singh, 2021*; *Høie et al., 2022*). The matrices presented here are, however, both cellular abundance- and structure context-specific. We provide an extensive analysis of the constructed matrices and use them to show that a considerable amount of the variation in the VAMP-seq data can be explained by taking only wild-type residue solvent accessibility and wild-type and variant residue types into account. Our results thereby highlight that simple biochemical considerations can compete with complex biophysics- and deep learning-based models for predicting the impact of residue substitutions on cellular abundance, although we note that predictions are in both cases non-perfect and that more work is needed before the abundance of protein variants can be accurately modelled. We use the derived substitution matrices to explore the relationship between variant abundance and thermodynamic stability and show that the matrices likely contain an abundance-specific signal. Finally, we also use the substitution matrices to propose a simple approach that, given a saturation mutagenesis dataset for the protein of interest, can be used to (*i*) assess structural models of the protein under the in vivo conditions used for the MAVE and (*ii*) discover solvent-exposed residues that play important roles for maintaining cellular protein levels.

## Results and discussion

### Summary and comparison of VAMP-seq datasets

To perform our large-scale analysis of substitution effects on protein abundance, we first collected previously published VAMP-seq data for six soluble human proteins, all known to be involved in human disease or drug metabolism. Although VAMP-seq has been applied to membrane proteins (*Chiasson et al., 2020*; *Amorosi et al., 2021*), residue substitutions might affect the cellular abundance of membrane and non-membrane proteins at least partially through different mechanisms (*Chiasson et al., 2020*), and we therefore limited our study to only include soluble proteins. Our dataset thus consisted of PTEN (*Matreyek et al., 2018*; *Matreyek et al., 2021*), TPMT (*Matreyek et al., 2018*), CYP2C9 (*Amorosi et al., 2021*), NUDT15 (*Suiter et al., 2020*), ASPA (*Grønbæk-Thygesen et al., 2024*), and PRKN (*Clausen et al., 2024*) VAMP-seq scores (*Supplementary file 1*). We realise that CYP2C9 is in fact anchored to the endoplasmic reticulum membrane through an N-terminal, trans-membrane helix, but the protein consists mostly of a large domain found in the cytoplasm (*Amorosi et al., 2021*), and we therefore included abundance scores for residues found in the cytoplasmic, soluble domain in our analysis.

The variant abundance scores for each of the proteins in our dataset were measured in individual VAMP-seq experiments, and the six experiments were all performed by sorting variant-expressing cells into four equally populated bins based on cellular fluorescence intensities. In all cases, the cellular abundance of a variant was quantified by taking a weighted average of the variant frequency across the four bins. The resulting scores were for each protein normalised so that a variant abundance score of 0 corresponds to nonsense-variant-like abundance and of 1 to wild-type-like abundance. We summarised and compared the six datasets by calculating a number of dataset descriptors and by mapping average abundance scores per position onto the protein structures (*Figure 1*). When combined, the six VAMP-seq datasets contain in total 31,614 single residue substitution variant abundance scores. The scores span the full lengths of the protein sequences, although the mutational depth, that is, the average number of substitutions per residue position with a reported score after quality filtering, varies considerably from protein to protein (*Figure 1B*). The mutational completeness of the datasets vary accordingly, with a minimum of 57% for PTEN and a maximum of 99% for PRKN (*Figure 1C*). As the proteins also have different sequence lengths (*Supplementary file 1*), they contribute unequally to the total pool of variants (*Figure 1D*).

We note that in spite of the similar experimental methodology and strategy for calculation of reported scores there are considerable differences between the shapes of the abundance score distributions obtained in the six experiments (*Figure 1—figure supplement 1*). PTEN, TPMT, and CYP2C9 have abundance score distributions with broad peaks at low and high abundance, whereas NUDT15, ASPA, and PRKN have almost binary distributions with narrow peaks, in particular for abundance scores close to 0. All distributions appear bimodal, with the exception of the CYP2C9 score distribution, which has an additional peak around 0.5. Low abundance scores are generally distributed around 0, but for PTEN, the low abundance-peak is shifted towards higher scores. The reported abundance score standard deviations also differ in magnitude across datasets (*Figure 1—figure supplement 2*). To represent the noise level in each dataset using a single number, we resampled the individual score distributions based on score standard deviations and calculated Pearson's correlation coefficient $r$ (and Spearman's rank correlation coefficient $r_s$) between the original datasets and the resampled datasets. In addition to indicating the dataset noise levels, the resulting $r$ (and $r_s$) values also provide an upper bound for how well abundance score predictions from a perfect variant abundance model would correlate with the experimental data (*Figure 1E*). Since dataset noise levels and the broadness of peaks in the score distributions appear correlated, we assume that the distribution shape differences originate at least partly from the varying noise levels in the datasets.

We hypothesise that the dissimilar score distributions also arise from a combination of intrinsic differences between the six protein systems and from method-related, assay-specific influences on the measured scores. The PTEN and TPMT score distribution differences have for example previously been suggested to reflect dissimilar wild-type protein melting temperatures (*Matreyek et al., 2018*). Additionally, the common VAMP-seq strategy to sort an equal number of variant-expressing cells into four bins has important implications, since variant effect scores produced in this manner are expected to depend on the composition of the variant library being sorted. Effectively, the input library composition determines the absolute values of the measured scores, and while the sorting and

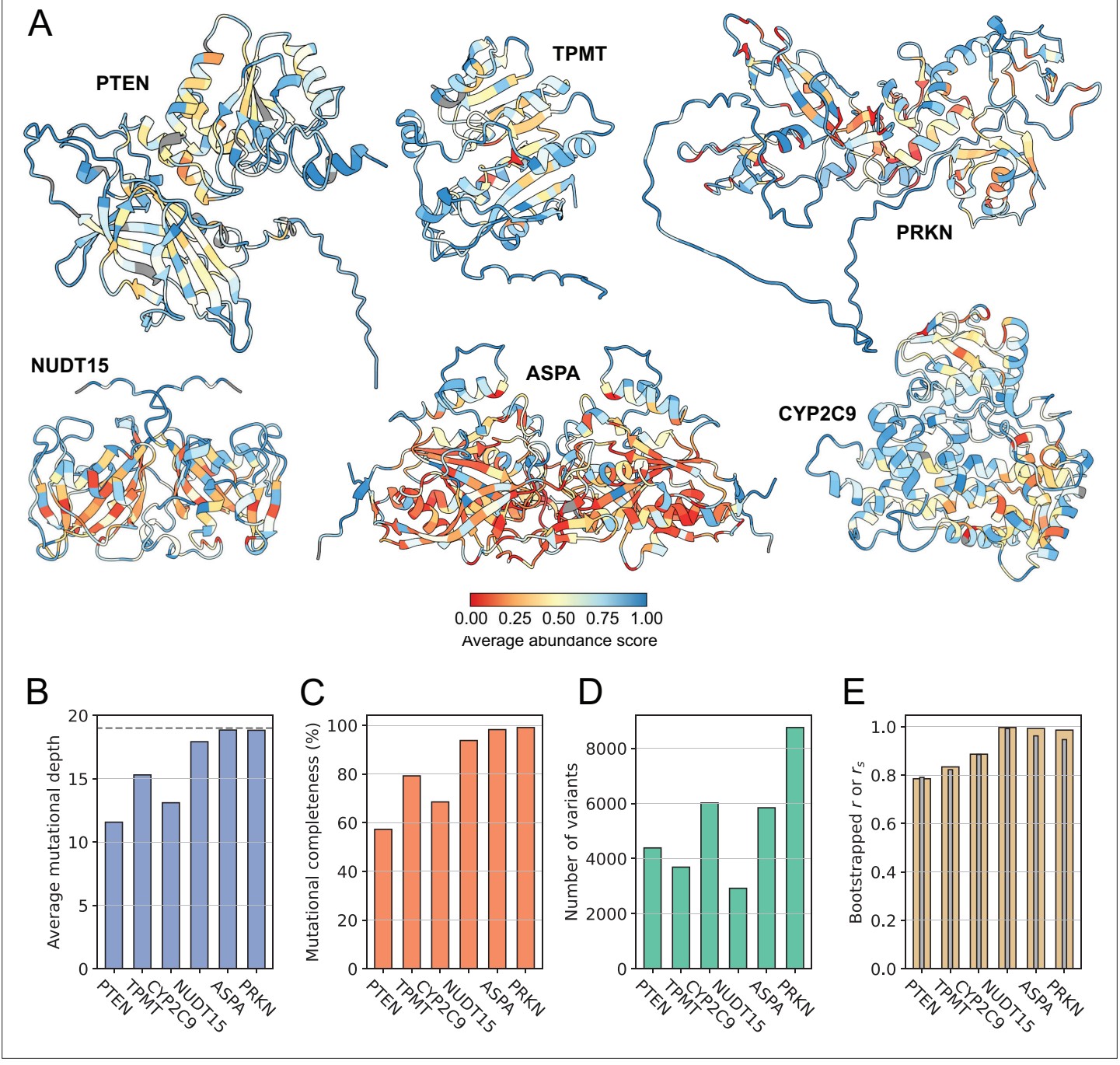

**Figure 1.** Overview of VAMP-seq datasets and proteins. (**A**) Structures of proteins previously studied with VAMP-seq coloured by average VAMP-seq abundance score per residue. Averages are shown for all residues with abundance scores available, even if only one or few scores have been reported for the residue. Homodimer structures are shown for NUDT15 and ASPA, and residues with no reported scores are shown in grey. (**B–E**) Overview of VAMP-seq dataset sizes, completenesses and noise levels, showing specifically (**B**) the average number of variant scores per residue position, with 19 (indicated with dashed line) being the highest possible value, (**C**) mutational completeness, that is, the percentage of all possible single residue substitution variants with an abundance score in the dataset, (**D**) the total number of single residue substitution abundance scores per protein, and (**E**) Pearson's correlation coefficient $r$ (yellow) and Spearman's rank correlation coefficient $r_s$ (grey) between originally reported and resampled abundance score datasets, with resampling based on reported abundance score standard deviations.

The online version of this article includes the following figure supplement(s) for figure 1:

**Figure supplement 1.** VAMP-seq abundance score distributions shown separately for each of the six datasets included in our analysis.

**Figure supplement 2.** VAMP-seq abundance score standard deviations plotted as function of abundance scores separately for each VAMP-seq dataset.

**Figure supplement 3.** Summary of protein structure and sequence compositions.

sequencing approach may produce an assay-independent variant ranking, the absolute values of the abundance scores are thus experiment-specific. In related work, we have expanded more on this point and explored how assay-specific influences may be accounted for when aggregating scores across datasets (*Schulze et al., 2025*).

In this study, we explore the six VAMP-seq datasets together without performing any additional within-dataset score transformations or across-dataset score normalisation. We show below that 'naively' combining VAMP-seq scores from the six experiments yields results that are largely in accordance with biochemical expectations and, importantly, facilitates useful structural analyses and construction of a relatively accurate prediction framework. We thus argue that direct combination of VAMP-seq scores across datasets is informative in spite of the methodological impact on absolute score values.

## Abundance scores correlate with the degree of residue solvent exposure

Our mapping of the average abundance score per residue onto the protein structures shows expected and previously described trends, most obviously that many solvent-exposed residues have high average abundance scores and that substitutions of core residues decrease abundance on average (*Figure 1A*). Exceptions to these patterns exist, as some solvent-exposed residues for instance appear intolerant to substitution. These exceptions have to some extent been described and analysed in the original VAMP-seq publications. We note that the average abundance scores differ in magnitude from protein to protein, in particular in core regions.

To quantify the relation between the structural context and mutational tolerance of a residue in the context of abundance, we used the wild-type structures of the six proteins, which have relatively similar structural compositions (*Figure 1—figure supplement 3*), to calculate features describing local residue environment, including relative solvent accessible surface area (rASA) and a weighted contact number (WCN). The WCN of a residue quantifies contact formation with all other residues in the protein so that a residue in a densely packed region will have a high WCN and vice versa. While rASA is sensitive to different residue environments on or close to the protein surface, rASA does not capture variation in the structure within the protein core. On the contrary, WCN has dynamic range for buried residues and is thus a complementary descriptor. Structural features were calculated with homodimer structures for NUDT15 and ASPA, in accordance with previous findings (*Suiter et al., 2020*; *Grønbæk-Thygesen et al., 2024*) and results described below. Monomeric structures were used for the four remaining proteins.

Both rASA and WCN correlate moderately well with the abundance scores, although the correlations are considerably higher for some proteins (PRKN and TPMT) than others (ASPA) (*Figure 2—figure supplement 1*, *Figure 2—figure supplement 2*). Generally, rASA captures the ranking of variant effects better than the WCN; Spearman's rank correlation coefficient, $r_s$, is on average 0.45 (0.54) between abundance scores and rASA and 0.41 (0.43) between abundance scores and WCN when including all datasets (or when including all except for ASPA) (*Figure 2—figure supplement 1*). When looking at the average abundance score per residue, we found solvent-exposed residues with reduced to strongly reduced abundance in all proteins, although in particular in ASPA and PRKN. Some residues buried in the structures on the other hand appear mutationally tolerant (*Figure 2—figure supplement 2*).

## Abundance score predictions from simple structural considerations

Based on the correlations between abundance scores and rASA and WCN, we hypothesised that combining the simple structural descriptors with information about the physicochemical properties of wild-type and variant residue types would allow us to explain the effects of many substitutions on cellular abundance (*Figure 2*). To test this hypothesis, we first established a baseline framework without using structural information, after which we extended our analysis to take structure into account.

For the structure-free analysis, we used the combined pool of abundance scores from all six proteins to construct a substitution matrix that, for each of the 380 possible amino acid residue substitution types, contains the abundance score average (*Figure 2—figure supplement 3*). We tested the ability of the substitution matrix to act as an abundance score predictor using leave-one-protein-out

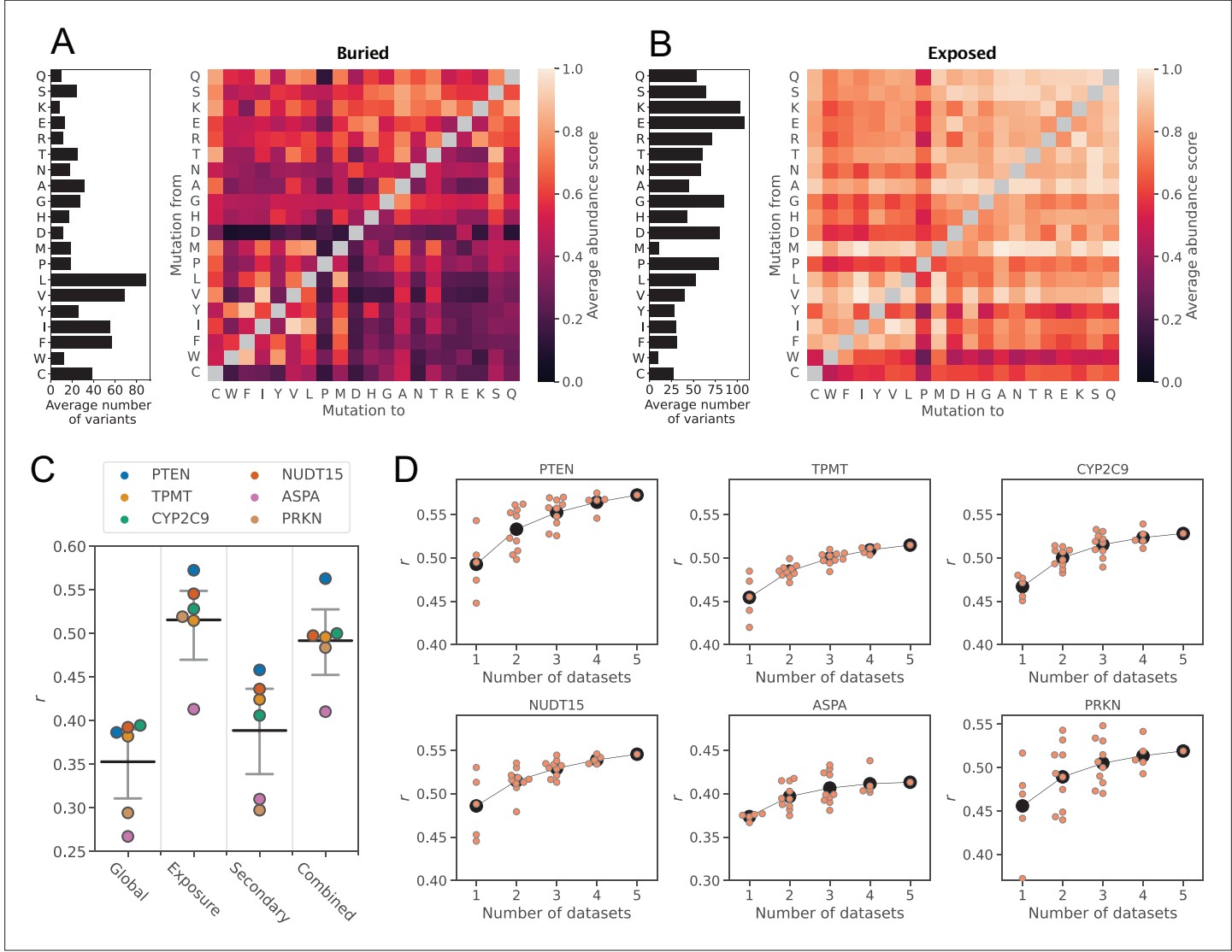

**Figure 2.** Average abundance score substitution matrices predict unseen abundance data. Average abundance scores for all possible amino acid residue substitution types for residues sitting in either (**A**) structurally buried or (**B**) solvent-exposed environments in the wild-type protein structures. The bar plot to the left of each substitution matrix indicates the average number of abundance scores that was used to calculate score averages in each row of the matrix. Amino acids are sorted according to row averages in the global substitution matrix (***Figure 2—figure supplement 3***). (**C**) Correlations between experimental abundance scores and abundance scores predicted from average abundance score substitution matrices using leave-one-protein-out cross-validation. The matrices used for predictions were constructed using all data (Global) and for different residue categories based on residue solvent-exposure (Exposure), secondary structure (Secondary), or both solvent-exposure and secondary structure (Combined). The horizontal, black bars mark the average correlation coefficient for each category, and the grey bars mark the 95% confidence intervals of the average $r$. Confidence intervals were estimated using bootstrapping by resampling the six $r$ values 10,000 times per model type. (**D**) Correlations between experimental abundance scores and abundance scores predicted from average abundance score substitution matrices for residues sitting in either buried or exposed environments. Scores were predicted for each protein (indicated above each plot) using average abundance score substitution matrices calculated using an increasing number of VAMP-seq datasets. Orange data points mark the prediction result from a single specific combination of datasets, and the average for each dataset count is indicated with black circles.

The online version of this article includes the following figure supplement(s) for figure 2:

**Figure supplement 1.** rASA and WCN correlate with single residue substitution abundance scores.

**Figure supplement 2.** rASA and WCN correlate with residue-averaged abundance scores.

**Figure supplement 3.** Average abundance score substitution matrices calculated using the six VAMP-seq datasets.

**Figure supplement 4.** VAMP-seq abundance scores plotted against abundance score predictions obtained from average abundance score substitution matrices constructed without consideration of the structural context of the wild-type residue (Global in ***Figure 2***).

*Figure 2 continued on next page*

*Figure 2 continued*

**Figure supplement 5.** VAMP-seq abundance scores plotted against abundance score predictions obtained from average abundance score substitution matrices constructed by considering the degree of solvent exposure of the wild-type residue (Exposure in *Figure 2*).

**Figure supplement 6.** VAMP-seq abundance scores plotted against abundance score predictions obtained from average abundance score substitution matrices constructed by considering the secondary structure context of the wild-type residue (Secondary in *Figure 2*).

**Figure supplement 7.** VAMP-seq abundance scores plotted against abundance score predictions obtained from average abundance score substitution matrices constructed by considering both the degree of solvent exposure and the secondary structure context of the wild-type residue (Combined in *Figure 2*).

**Figure supplement 8.** Comparison of average abundance score substitution matrices across proteins and residue environments.

**Figure supplement 9.** Correlations between experimental abundance scores and abundance scores predicted from the six average abundance score substitution matrices that consider both residue burial and secondary structure context of the wild-type residue.

**Figure supplement 10.** Grid search for selection of structure feature cutoffs to classify residues as solvent-exposed or structurally buried.

cross-validation; that is, we recalculated the matrix leaving out the entire VAMP-seq dataset of a single protein and then used the averages in the recalculated matrix as abundance score predictions for substitutions in the omitted protein. While the average abundance scores are not particularly good predictors of variant abundance (*Figure 2C*, *Figure 2—figure supplement 4*), some signal in the experimental data is indeed captured even without considering the structural context of the wild-type residues (*Figure 2—figure supplement 4*). Previous studies have reached similar conclusions in analyses that calculate structure-independent substitution matrices and predict variant fitness by combining data from a variety of MAVE experiments (*Munro and Singh, 2021*; *Høie et al., 2022*).

We next explored whether we could improve upon this baseline by taking simple structural features into account. We used rASA values calculated for the wild-type protein structures to classify all residues as either solvent-exposed or structurally buried. We then calculated two new substitution matrices, in one matrix averaging over variant abundance scores from residues belonging to the solvent-exposed class and in the other over scores from buried residues (*Figure 2A and B*). Using the same approach as above, we evaluated how well our exposure-based substitution matrices could predict unseen abundance data using leave-one-protein-out cross-validation. The abundance score prediction of a variant was thus obtained by looking up the abundance score average of the relevant substitution type in either the exposed or buried residue matrix depending on the structural class of the residue for which a prediction was made. Accounting for the structural context of the wild-type residue considerably increases the ability of the model to capture the impact of substitutions on abundance, with the average *r* across datasets increasing from 0.35 to 0.52 when not only the substitution type but also the degree of residue solvent exposure is considered (*Figure 2C*, *Figure 2—figure supplement 5*).

We next tested if additional substitution effects could be captured by including secondary structure in our framework. To perform this analysis, we calculated two new sets of substitution matrices. In the first set, we averaged abundance scores for residues in different types of secondary structure and obtained three matrices with average effects in helix, strand, and loop structure (*Figure 2—figure supplement 3*). To create the other set of matrices, we split residues by both secondary structural class and degree of solvent exposure, creating residue classes such as 'solvent-exposed helix' and 'buried strand', which resulted in six matrices of average effects. We performed abundance score predictions with the two sets of matrices using the cross-validation approach described above. The substitution matrices based exclusively on residue secondary structure perform only slightly better in the prediction task than the structure-independent matrix (*Figure 2C*, *Figure 2—figure supplement 6*), and secondary structure is by itself thus not a suited descriptor to capture variant effects on abundance. The combination of solvent-exposure and secondary structure type similarly does not improve predictions compared to using only solvent exposure (*Figure 2C*, *Figure 2—figure supplement 7*).

Since our VAMP-seq dataset is relatively small and limited to six proteins, we next assessed if our substitution matrix-based abundance models could be expected to improve with the addition of more VAMP-seq data in the future. For this assessment, we recalculated all exposure-based matrices using all possible combinations of the six datasets. We then performed predictions again, for every protein using all matrices in which the data for this protein did not occur to perform predictions. In this way, we estimated how *r* varied with the number of datasets included in the substitution matrix calculation.

While using more datasets improves predictions on average, we observe that, depending on the exact combination of datasets, predictions can be as good or better when only one or few datasets are used to calculate the abundance score averages as when five datasets are used (*Figure 2D*). Some datasets thus appear to be more related than others (*Figure 2—figure supplement 8*). The average *r* seems to be plateauing for all six proteins, suggesting that this framework would likely only improve to a small degree with the addition of more data in the future. We observed similar trends when we performed the same analysis for the Combined matrices (*Figure 2—figure supplement 9*).

The results presented above show that combining simple structural features with information about the wild-type and variant residue types enables us to capture the effects of many substitutions on the cellular abundance of a protein. It is at the same time clear that some substitution effects are not properly captured within this prediction framework. We in particular find one common group of outliers in predictions of the NUDT15, ASPA, and PRKN datasets, namely substitutions that experimentally have strongly reduced abundance scores, but which the substitution matrices predict to have wild-type-like or nearly wild-type-like abundance (*Figure 2—figure supplements 4–7*). These prediction outliers likely occur for several reasons, including due to (*i*) biochemical and biophysical features not captured in the substitution matrix framework and (*ii*) differences between VAMP-seq datasets (*Figure 1—figure supplement 1*) that occur for technical rather than biochemical reasons (*Schulze et al., 2025*). An accurate analysis of the prediction outliers, for example to inform model refinement, would thus involve disentangling these two potential outlier causes, which, in addition to the substantial noise levels of the data, makes outlier detection and analyses challenging.

## Substitution matrices and ΔΔG predictors rank variant effects equally well

We and others have previously observed a correlation between the cellular abundance of variants and (predicted) differences between variant and wild-type protein folding stabilities $(\Delta\Delta G = \Delta G_{\text{variant}} - \Delta G_{\text{wild type}})$, both in low-throughput cellular studies (*Nielsen et al., 2017*; *Scheller et al., 2019*; *Abildgaard et al., 2019*) and in high-throughput VAMP-seq studies (*Matreyek et al., 2018*; *Suiter et al., 2020*; *Cagiada et al., 2021*; *Høie et al., 2022*; *Gerasimavicius et al., 2023*; *Grønbæk-Thygesen et al., 2024*; *Clausen et al., 2024*). Thermodynamically destabilised protein variants may be more likely to populate unfolded conformations than the wild-type protein, thereby exposing hydrophobic core residues that can trigger the cellular protein quality control (PQC) system to degrade the protein, for example to avoid aggregate formation (*Hipp et al., 2019*; *Clausen et al., 2019*). While ΔΔG values usually correlate with cellular abundance, we do not necessarily expect a perfect (rank) correlation between the two quantities, since substitutions of single residues can trigger degradation via a number of mechanisms that do not require complete unfolding from the native state. For example, a substitution can introduce neo-degrons at surface-exposed positions (*Clausen et al., 2024*), cause local unfolding that exposes a degron (*Kampmeyer et al., 2022*) or change degradation-relevant post-translational modification patterns (*Vazquez et al., 2000*). For one of several possible reasons, a variant associated with a small ΔΔG might thus be completely degraded in the cell.

To explore the relationship between variant abundance and folding stability, and to assess the performance of our substitution matrix-based prediction method, we calculated ΔΔG scores for all single residue substitution variants of the six VAMP-seq proteins. We predicted ΔΔG scores with three different models, the Rosetta energy function (*Park et al., 2016*) and the deep learning-based RaSP (*Blaabjerg et al., 2023*) and ThermoMPNN (*Dieckhaus et al., 2024*) models. We calculated $r_s$ between ΔΔG predictions and VAMP-seq scores, analysing scores from each protein separately. As expected and observed in previous work, we find that variant ΔΔG values correlate with abundance scores (*Figure 3A*), although with noticeable differences in the magnitude of the rank correlation coefficient across datasets; for example, $r_s$ ranges from –0.45 for PRKN to –0.59 for NUDT15 when predicted with Rosetta and from –0.51 for TPMT to –0.65 for NUDT15 when predicted with ThermoMPNN (*Figure 3A*). While an $r_s$ of –1.0 is not expected due to noise in the VAMP-seq data (*Figure 1E*), it is clear that predicted ΔΔG values nevertheless cannot fully explain the abundance data, likely due to a combination of limited ΔΔG prediction accuracy (*Blaabjerg et al., 2023*; *Dieckhaus et al., 2024*; *Degn et al., 2025*) and the fact that VAMP-seq abundance measurements contain other contributions than global changes in protein stability (ΔΔG). Moreover, variation in $r_s$ across the six systems possibly

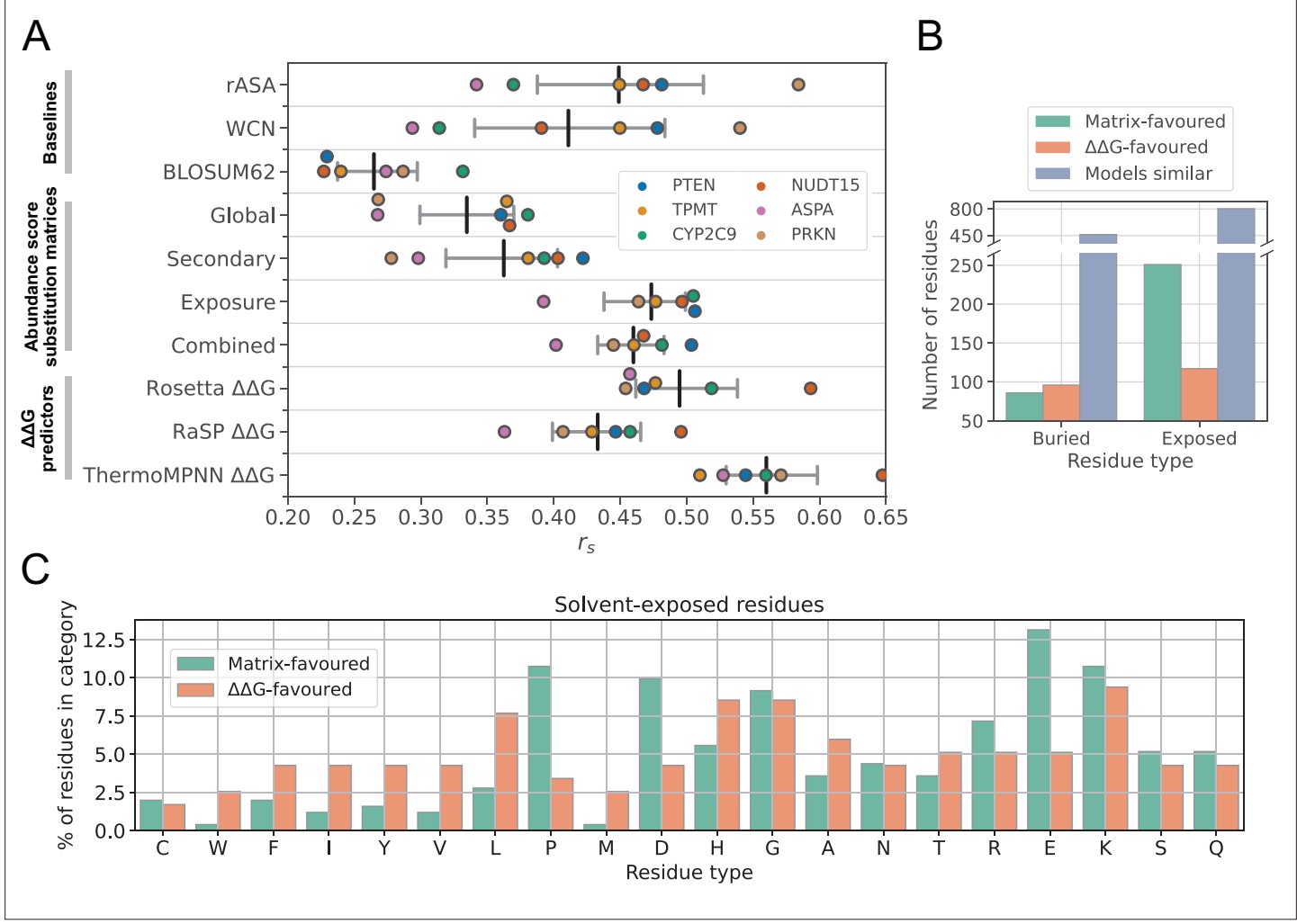

**Figure 3.** Ranking of variant abundance by substitution matrices and ΔΔG predictors. (**A**) $r_s$ between experimental abundance scores and variant effect score predictions from a number of models, including abundance score substitution matrices (Global, Secondary, Exposure, and Combined) and ΔΔG predictors (Rosetta, RaSP, and ThermoMPNN). $r_s$ between experimental abundance scores and residue rASA or WCN values are also included, as well as variant effect scores predicted with the BLOSUM62 matrix (*Henikoff and Henikoff, 1992*). For the three ΔΔG predictors and WCN values, absolute $r_s$ values are shown. Coloured points indicate $r_s$ for individual proteins. Vertical bars in black mark the average $r_s$ across the six proteins, and grey bars indicate 95% confidence intervals of the $r_s$ average. Confidence intervals were estimated using bootstrapping by resampling the six (one per protein) rank correlation coefficients 10,000 times for every predictor. (**B**) Number of individual residue substitution profiles captured best by the exposure-based substitution matrices (green) or ThermoMPNN ΔΔG predictions (orange), shown separately for buried and exposed residues. Variant abundance is often ranked better by the substitution matrices than ΔΔG values for exposed residues. For most residues, the substitution matrix and ΔΔG models rank residue-level variant effects with similar accuracy (blue). (**C**) Distributions of amino acid residue types in the groups of matrix-favoured (green) and ΔΔG-favoured (orange) solvent-exposed residues. For every amino acid residue type, the residue count in either the matrix- or ΔΔG-favoured category was normalised to the total number of residues in the category, thus revealing enrichment and depletion of certain amino acid types in one of the two groups compared to in the other.

The online version of this article includes the following figure supplement(s) for figure 3:

**Figure supplement 1.** Predicted ΔΔG values correlate with predicted abundance scores.

**Figure supplement 2.** Abundance score substitution matrices for variants of solvent-exposed residues classified as either (**A**) matrix-favoured or (**B**) ΔΔG-favoured, as well as (**C**) the difference between these two substitution matrices.

indicates that effects of substitutions on abundance are dominated by ΔΔG effects to a higher degree in some systems than in others.

Abundance scores are ranked best by ThermoMPNN with an average $r_s$ of –0.56 across the six proteins, in line with observations that ThermoMPNN captures experimentally measured stability changes more accurately than Rosetta and RaSP (*Dieckhaus et al., 2024*; *Degn et al., 2025*). We

note that all of the models we have tested, including ThermoMPNN, are far away from predicting abundance scores perfectly, even considering the $r_s$ upper limits given by the noise in the VAMP-seq data (*Figure 1E*). Rosetta and RaSP (the latter of which was trained to predict Rosetta scores rapidly) are far more complex models than any of the abundance score substitution matrices, and it is thus noteworthy that the exposure-based matrices have a predictive performance comparable to that of Rosetta or RaSP (*Figure 3A*). As an abundance score predictor, the exposure-based substitution matrices are thus competitive with RaSP, a deep learning model using a 3D convolutional neural network to process structural information, and Rosetta, which relies on a complex energy function and includes post-mutation structural sampling as part of the ΔΔG calculation pipeline. Based on this, we propose our matrix-based predictions, alongside ThermoMPNN, as a strong baseline for future abundance model development. We also emphasise the utility of simple, interpretable structure features for stability-related variant effect predictions.

## Abundance score predictions and ΔΔGs contain (some) orthogonal information

Variant ΔΔG predictions and substitution matrix-based abundance score predictions correlate only moderately well (*Figure 3—figure supplement 1*), suggesting that the two model types may achieve relatively similar rank correlations with the VAMP-seq data for at least somewhat different reasons. Analysis of the differences in model behaviour might shed light on the relationship between variant folding stability and cellular abundance and potentially guide improved abundance model design. We therefore next compared the exposure-based substitution matrices to the predicted ΔΔG values in detail, focusing the analysis on ThermoMPNN ΔΔG predictions. Below, we define the substitution profile of a residue as the vector of abundance scores for variants of that residue.

We compared ΔΔG predictions to predictions derived from the exposure-based substitution matrices by identifying all individual residues for which one model predicted the experimental abundance score substitution profile considerably better than the other model. For every individual residue in a protein, we specifically compared $r_{s,\Delta\Delta G}$ and $r_{s,\text{matrix}}$, which are the rank correlation coefficients between the experimental residue substitution profile and the ΔΔG-based ($r_{s,\Delta\Delta G}$) or the matrix-based ($r_{s,\text{matrix}}$) substitution profile predictions. To account for noise in the VAMP-seq data, we resampled the abundance scores of every individual substitution profile 10,000 times based on the experimental abundance score standard deviations. We then used the resampled profiles to ask how often $-r_{s,\Delta\Delta G} > r_{s,\text{matrix}}$ or $-r_{s,\Delta\Delta G} < r_{s,\text{matrix}}$ for a given residue. If one method consistently predicted the substitution profile of a residue better than the other (that is, in more than 95% of cases), we classified the residue either as "ΔΔG-favoured" (for $-r_{s,\Delta\Delta G} > r_{s,\text{matrix}}$) or "matrix-favoured" (for $-r_{s,\Delta\Delta G} < r_{s,\text{matrix}}$).

We used this approach to analyse 1816 residue substitution profiles distributed across the six proteins. Of the 1816 analysed residues, we identified 337 (19%) matrix-favoured residues and 213 (12%) ΔΔG-favoured residues. 1266 of the 1816 residues (70%) have substitution profiles that are not predicted considerably better by one method than by the other; for these residues, $r_{s,\Delta\Delta G}$ and $r_{s,\text{matrix}}$ cannot be told apart, either because the two predictors agree on how variants should be ranked or due to noise in the experimental data. When considering buried and exposed residues separately, we find 251 of 1172 (21%) exposed residues to be matrix-favoured, while only 117 of 1172 (10%) exposed residues are ΔΔG-favoured (*Figure 3B*). The substitution profiles of exposed residues thus tend to be captured best by the substitution matrix model or equally well by the two model types. We also identify 182 of 644 (28%) buried residues for which either ΔΔG values or substitution matrix-based scores most accurately rank variant effects. For buried residues, the difference in the number of ΔΔG- and matrix-favoured residues is small (*Figure 3B*). Since most substitution profiles have indistinguishable $r_{s,\Delta\Delta G}$ and $r_{s,\text{matrix}}$ values, we do not expect a model that combines substitution matrix-based and ThermoMPNN predictions to outperform the $r_s$ of ThermoMPNN when evaluated globally, that is, across all variants in the individual systems.

We next set out to characterise the residues that we had classified as either matrix- or ΔΔG-favoured. To this end, we first analysed the amino acid composition of the matrix-favoured and ΔΔG-favoured residue groups, focusing our analysis on solvent-exposed positions. We specifically counted every amino acid type in the group of solvent-exposed and matrix-favoured residues (or in the group of solvent-exposed and ΔΔG-favoured residues) and then normalised the counts to the total number

of residues in the group (*Figure 3C*). This analysis reveals that certain amino acid types are enriched in the matrix-favoured residue group compared to in the ΔΔG-favoured group, and vice versa. Most notably, we find that aspartate, glutamate and proline are particularly enriched in the matrix-favoured residue group, meaning that for these three amino acid types, the substitution matrix model captures variant effects particularly well. The ΔΔG-favoured group is, on the contrary, enriched in aromatic and several non-polar amino acid types, and so the variant abundance of solvent-exposed residues of these amino acid types tend to be captured best by ThermoMPNN ΔΔG scores. We note that there is a relatively small number of abundance scores for substitutions from solvent-exposed aromatic and non-polar residues in our combined VAMP-seq dataset (*Figure 2B*), and so the latter result might reflect that the substitution matrix for exposed residues has not seen a lot of training data for aromatic and non-polar amino acid types.

We further asked how the substitution profiles of solvent-exposed matrix-favoured and ΔΔG-favoured residues differ. To answer this question, we calculated average abundance score substitution matrices using VAMP-seq data for all residues in the two residue categories (*Figure 3—figure supplement 2A and B*) and then determined the difference between the two resulting substitution matrices (*Figure 3—figure supplement 2C*). We find that the average abundance scores of substitutions from several charged and polar residues to aromatic or non-polar residues tend to be smaller for matrix-favoured residues than for ΔΔG-favoured residues. Assuming that ThermoMPNN accurately models variant ΔΔG, these results suggest that non-conservative substitutions of certain polar or charged residues on the protein surface (namely those belonging to the matrix-favoured residue group) may reduce cellular abundance more than what would be expected from a purely thermodynamic perspective, and these effects seem to be captured in the substitution matrix model.

In conclusion, we have shown that predicted ΔΔG values and entries in the substitution matrices are not identical with respect to variant effect ranking, neither for buried nor exposed residues, suggesting that the substitution matrices may contain abundance-specific signal. It is interesting that exposed residues show a tendency to be matrix-favoured, as this is in line with expectations that some surface-level substitutions may affect folding stability and interactions with the PQC system in different ways. Since neither the substitution matrices nor ThermoMPNN are perfect models, it is, however, difficult to very confidently conclude that discrepancies between predictions from the two different models necessarily indicate that the effects of certain variants on abundance are driven exclusively by folding free energy changes while other variant effects are products of more complex reactions in the cell.

## Substitution matrices reveal variant effect trends across protein backgrounds

We find that the average substitution effects in the VAMP-seq-derived substitution matrices are in general in accordance with biochemically expected and previously observed patterns (*Munro and Singh, 2021*; *Høie et al., 2022*). Across residue environments, substitutions conserving the physicochemical properties of the reference residue are often tolerated well, or at least better than non-conservative substitutions, and it is moreover clear that residues found in structurally buried environments tolerate substitutions less than solvent-exposed residues (*Figure 2A and B*). While substitution of residues in buried environments likely decreases protein cellular abundance on average because the thermodynamic stability is compromised, substitution of exposed residues might affect cellular stability through different mechanisms (*Correa Marrero and Barrio-Hernandez, 2021*), for instance by interference with formation of intermolecular interactions (and thus potentially thermodynamic stability of complexes) (*Suiter et al., 2020*; *Grønbæk-Thygesen et al., 2024*) or through modification of exposed degradation motifs (*Clausen et al., 2024*).

We observe that, at solvent-exposed sites, substitutions from polar or charged residues to non-polar or aromatic residues are not tolerated as well as conservative substitutions. Comparing matrices across residue environments shows that substitution to proline is, as expected and previously observed, not tolerated well in any environment, and substitution from proline is also often disruptive. Mutation of cysteine to any other residue also appears to on average reduce abundance considerably in several environments. However, leaving out PRKN abundance scores from the calculation of the averages makes mutation from cysteine much less disruptive (*Figure 4—figure supplement 1*), and the average effects when all proteins are included are thus dominated by the $Zn^{2+}$-coordinating

cysteine residues that are critical for the abundance of PRKN (*Clausen et al., 2024*). Generally, substitution matrices constructed with individual datasets display unique features and relate to each other more closely for certain protein pairs than others (*Figure 2—figure supplement 8A and B*), while leave-one-protein-out substitution matrices resemble each other relatively closely (*Figure 2—figure supplement 8C and D*).

In line with expectations and previous results (*Munro and Singh, 2021*; *Høie et al., 2022*), the global, structure-independent matrix is furthermore asymmetric (*Figure 4—figure supplement 2*), that is, for many residue pairs, substitution from residue *X* to *Y* tends to affect abundance differently than substitution from residue *Y* to *X*. In the global context, substitutions from non-polar or aromatic residues to polar or charged residues for example tend to be more disruptive than the opposite substitutions. Such asymmetry is, per construction, not present in traditional substitution matrices used for construction of sequence alignments (*Dayhoff et al., 1978*; *Henikoff and Henikoff, 1992*), although more recent work has derived asymmetric matrices using sequence data (*Dang et al., 2022*). The matrix with average abundance scores from buried residues is also asymmetric, in spite of the fact that substitutions from polar or charged residues to hydrophobic or aromatic residues (and not only the opposite) tend to be quite disruptive of abundance in buried environments (*Figure 4—figure supplement 2*). While some asymmetric features are still present when only solvent-exposed residues are considered, the asymmetry is less pronounced, meaning that non-conservative substitutions often affect cellular stability relatively similarly irrespective of the biochemistry of the wild-type residue in this structural environment (*Figure 4—figure supplement 2*).

From visual inspection of the substitution matrices, it is clear that certain pairs and groups of amino acid types share a number of substitution profile features. In a given environment, each amino acid type has two substitution profiles in form of the two vectors of nineteen abundance score averages describing effects of substitution to or from the given amino acid. We explored substitution profile similarities by performing hierarchical clustering of the average score matrices, focusing our analysis on effects in buried and exposed environments (*Figure 4A*). The analysis of buried residues identifies several clusters that are in accordance with residue biochemistry; for example, when considering mutational profiles for substitutions *to* a given residue, one cluster contains the aromatic residues, another cluster consists of several non-polar residues, and residues with similar charges are identified as closely related. We also observed that, in buried environments, substitutions to glycine are in many cases as detrimental as substitutions to polar or charged residues. The clustering also highlights that substitutions from non-polar residues to the aromatic residues are not well-tolerated on average, potentially due to differences in the volume of the side chain. In fact, substitutions from non-polar buried residues to the aromatic residues share features with substitutions from non-polar buried residues to charged or polar residues. In the exposed environment, we also found clusters that are generally in accordance with residue biochemistry.

For both buried and solvent-exposed environments, several residue groups identified in the clustering of the matrix rows are also identified in the clustering of the matrix columns. However, the dendrograms along the two axes are not identical, neither in the buried nor in the exposed context. The order in which clusters form depends on the axis, and some residues appear in different clusters depending on whether they are being mutated from or to. For example, serine cluster with the majority of polar and charged residues in the *mutation from* direction in the buried matrix, but not in the *mutation to* direction. More surprisingly, aspartate and glutamate behave similarly when they occur as the variant residue in the buried environment, but have more different mutational patterns when they are the wild-type residue. Generally, buried aspartate appears to play a critical role for maintaining abundance and is clearly an outlier in the *mutation from* direction, as it does not seem to cluster with any other residues. Even mutation from aspartate to glutamate is not tolerated well in the protein core, highlighting that the importance of buried aspartate for cellular stability is not only charge-related. In the exposed environment, aspartate and glutamate do cluster together as both wild-type and variant residues.

We further quantified the relations between the average substitution profiles of the 20 amino acids using principal component analysis (PCA). We concatenated the 20 *mutation to* and 20 *mutation from* abundance score averages for every amino acid residue type and used the 20 resulting 40-vectors as input for the PCA. In that way, the PCA explored amino acid similarities and differences considering simultaneously the effects of substitution from and to a given amino acid. For buried residues, the

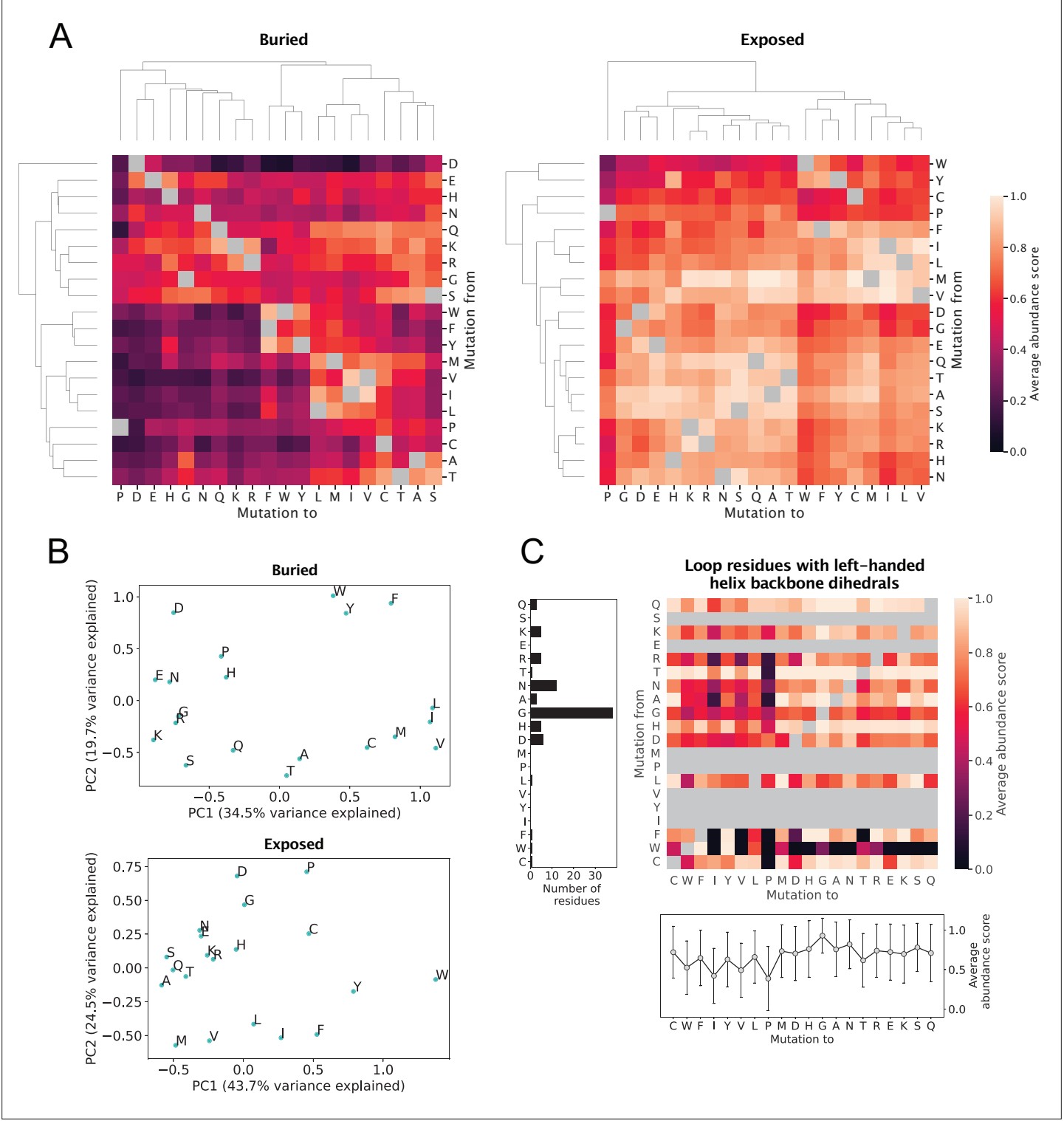

**Figure 4.** Analysis of average abundance score substitution matrices. (**A**) Hierarchical clustering of average abundance score substitution matrices for buried and solvent-exposed environments. Clustering was performed along both axes of the matrices. Grey squares indicate a synonymous substitution. (**B**) Principal component analysis of substitution profiles describing the effects of all substitutions from and to each of the 20 amino acid residues. The analysis was performed using average abundance scores for residues in buried and exposed environments separately. (**C**) Analysis of substitution profiles of loop residues with backbone dihedral angles similar to those found in left-handed helices. The heat map shows average abundance scores for all loop residues with left-handed helix-like backbone dihedral angles in the six proteins. Grey squares indicate a synonymous substitution as well as no data. The plot below the heat map shows average abundance scores for substitutions to each of the 20 amino acid residues for this particular class

*Figure 4 continued on next page*

*Figure 4 continued*

of loop residues, with error bars indicating the standard deviation over all abundance scores for the substitution type. The left bar plot shows the total number of loop residues that for each residue type was found across the six proteins to adopt the left-handed helix-like backbone conformation.

The online version of this article includes the following figure supplement(s) for figure 4:

**Figure supplement 1.** Average abundance score substitution matrices for (**A**) buried, (**B**) exposed and (**C**) loop environments calculated without abundance scores from the PRKN VAMP-seq dataset, but including all data from the five remaining datasets.

**Figure supplement 2.** Heatmaps indicating the degree of symmetry across the diagonal in average abundance score substitution matrices.

**Figure supplement 3.** Analysis of the impact of amino acid helix propensity on average abundance scores.

first principal component (PC1) appears to separate the polar and charged amino acids from the non-polar and aromatic amino acids (*Figure 4B*), showing (as expected) that amino acid hydrophobicity is an important property for explaining variation in the substitution profiles in buried environments. Along PC1, proline and glycine are placed close to the polar and charged amino acids. PC2 for buried residues clearly separates the nonpolar and aromatic residues, with most polar residues found at a shorter distance to the nonpolar residues than the aromatics. The PCA of average abundance scores in exposed environments separates the nonpolar and aromatic amino acids from the remaining amino acids along PC2 rather than PC1. The PC1 for residues in an exposed environment, which on its own explains a relatively high amount of variation in the substitution profiles, instead seems at least partially correlated with how well substitutions to polar and charged residues are tolerated (*Figure 4B*).

For residues in loops, we noticed that glycine, histidine, asparagine, and aspartate share certain substitution profile features, as they all appear particularly sensitive to mutation to $\beta$-branched residues, including threonine, isoleucine, and valine, as well as to tryptophan and proline (*Figure 2— figure supplement 3*). In loop structure, many residues, and in particular exactly glycine, histidine, asparagine, and aspartate, are able to adopt backbone $\phi$ and $\psi$ angles in the $0° < \phi < 180°$, $-90° < \psi < 90°$ area of the Ramachandran plot (*Hovmöller et al., 2002*). Backbone dihedral angles in this region of the Ramachandran plot are associated with left-handed helix formation, although the vast majority of residues adopting these dihedral angles do not actually take part in forming a left-handed helix (*Hovmöller et al., 2002*). A number of residue types, including isoleucine, threonine, proline, and valine, have very little density in this region of the Ramachandran plot (*Hovmöller et al., 2002*). We therefore hypothesised that the ability to adopt the $\phi$ and $\psi$ angles of left-handed helical conformations influences the average abundance scores of residues in loop structure. To test our hypothesis, we identified all loop residues with backbone angles in the $\phi,\psi$ area stated above. As expected, many of the identified residues were glycine and asparagine residues, whereas no isoleucine or valine residues in our six protein dataset were found to have this backbone conformation (*Figure 4C*). Moreover, for several residue types, we identified only a single residue or no residues with this conformation. We next calculated an average abundance score substitution matrix for the identified residues and averaged the abundance scores for substitutions to each of the 20 amino acids across reference residue types (*Figure 4C*). Clearly, substitution to proline, isoleucine, valine, and tryptophan are the least favourable substitution types on average, and substitutions to glycine are the most tolerated. Our analysis thus suggests that substitution of loop residues with left-handed helical backbone conformations typically disrupts cellular abundance if this conformation is not usual for the variant residue to adopt.

Finally, we tested whether known helix propensity scales could explain patterns in the abundance substitution matrices. To do so, we calculated the correlation between average abundance scores for residues found in $\alpha$-helices and different experimentally and statistically derived residue helix propensity scales (*Pace and Scholtz, 1998*). All correlations between average abundance scores and helix propensity scales are weak (*Figure 4—figure supplement 3*), indicating that the average substitution patterns in the abundance data are not dominated by helix propensities. A similar observation was made in a study aggregating data from a broader range of MAVEs (*Gray et al., 2017*).

## Average substitution profiles identify stabilising dimer interfaces

In vitro studies have demonstrated that NUDT15 (*Carter et al., 2015*) and ASPA (*Moore et al., 2003*; *Bitto et al., 2007*; *Le Coq et al., 2008*) form homodimeric assemblies in solution and when crystallised. In a previous analysis of the NUDT15 VAMP-seq dataset, the homodimer interface present in the

NUDT15 crystal structure was found to contain several residue sites at which the mutational tolerance was lower than the average tolerance across all residues and substitutions hence consistently resulted in decreased abundance (*Suiter et al., 2020*). Sensitivity towards substitutions has also been noticed for residues sitting in the monomer-monomer interface found in the ASPA crystal structure (*Grønbæk-Thygesen et al., 2024*). Since prior work thus suggests that NUDT15 and ASPA homodimerisation is important for maintaining cellular abundance of the proteins, we have used the NUDT15 and ASPA homodimer structures for all analyses described above. However, no quantitative analysis of the mutational tolerance and hence importance for maintaining cellular stability has actually been reported for the interface residues in ASPA, and so to justify our choice, we (re)analysed the crystal structure interfaces of the two proteins using the pooled VAMP-seq dataset.

We specifically examined whether NUDT15 and ASPA homodimerisation might affect the in vivo stability of the proteins in the VAMP-seq experiments by quantifying the extent to which the abundance scores of the homodimer interface residues resemble the average substitution scores of buried or solvent-exposed residues. For the analysis, we recalculated the exposure-based substitution matrices using only monomeric structures for the residue classification and leaving out either NUDT15 or ASPA data from the calculation. For every interface residue in the dimer structures, we then calculated the root-mean-square-deviation (RMSD) between all variant abundance scores reported for the interface residue and the average abundance score substitution profile for the interface residue amino acid type in either buried or exposed environments. For all interface residues, we thus obtained two numbers, $RMSD_{exposed}$ and $RMSD_{buried}$, which respectively quantify how closely related the substitution profile scores of the residue are to the average scores of exposed and buried sites (*Figure 5*). If the $RMSD_{buried}$ of a residue is smaller than the $RMSD_{exposed}$, we thus expect the residue to be buried in the monomer-monomer interface in the cell, and vice versa.

In NUDT15, the abundance scores for 11 out of 13 crystal structure interface residues are more similar to the average abundance scores of buried residues than exposed residues (*Figure 5A*). The disruption of the interactions formed by these 11 residues thus reduces the cellular abundance of NUDT15 relatively similarly to how disruption of the interactions of core residues would, suggesting that the 11 residues are buried in the dimer interface inside the cell. Residues L153 and L156 have abundance scores that resemble those of exposed residues more closely than those of buried residues. In the crystal structure of the dimeric assembly of the protein, L153 and L156 are positioned next to each other on the border of the dimer interface (*Figure 5C*). While the side chains of the two residues do point into the interface in the static structure, our analysis suggests that they are not more important for the cellular stability of NUDT15 than if they had been sitting in solvent-exposed positions. We note that this result is more clear for residue L153 than residue L156.

Using the same method, we found that 10 out of 14 interface residues in the ASPA dimer have a smaller $RMSD_{buried}$ than $RMSD_{exposed}$ (*Figure 5B*). The majority of the ASPA dimer interface residues thus appear structurally buried, and we conclude that ASPA exists as a dimer in vivo and that dimerisation is important for maintaining the cellular stability of the protein. In our analysis, we found crystal structure interface residues L25, F29, C61, and L265 to have abundance scores most similar to those of solvent-exposed residues of the same amino acid types. The four residues do not appear to be as critical for ASPA dimerisation and cellular stability as the rest of the interface residues. In the crystal structure, L265 is positioned in a loop at the border of the interface (*Figure 5D*), while L25, F29, and C61 are spatially close to each other at a site distant from L265 (*Figure 5E*). We note that for ASPA, the $RMSD_{buried}$ for several residues is relatively high although still smaller than $RMSD_{exposed}$, in accordance with the fact that the ASPA VAMP-seq dataset contains many variants with very low abundance.

## Discovering solvent-exposed residues important for cellular stability

Having analysed residues found in the monomer-monomer interfaces of the NUDT15 and ASPA structures, we next tested whether our method was able to identify other solvent-exposed residues with $RMSD_{buried} < RMSD_{exposed}$ on the non-interface surfaces of NUDT15 and ASPA as well as on the surfaces of the four other proteins in our dataset. At the same time, we tested whether any buried residues have $RMSD_{exposed} < RMSD_{buried}$. For the analysis, we used average abundance score matrices for buried and exposed residues calculated using only monomer structures and excluding data from the protein being analysed. For every residue in each of the six proteins, we calculated $RMSD_{exposed}$ and $RMSD_{buried}$ using all reported variant abundance scores for the residue (*Figure 5—figure supplement*

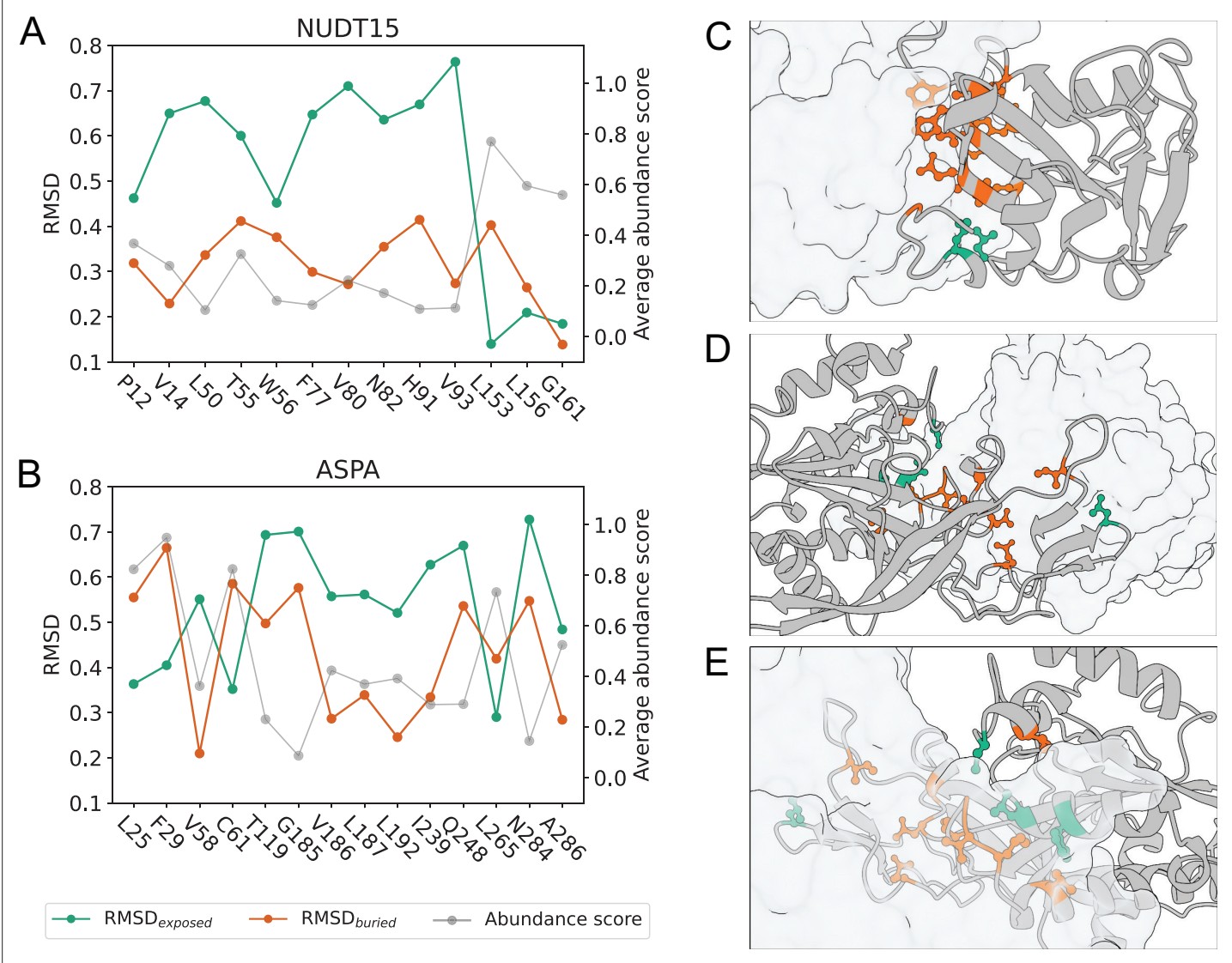

**Figure 5.** Homodimerisation stabilises NUDT15 and ASPA in vivo. RMSD between abundance score substitution profile and average abundance scores in buried (orange) and exposed (green) environments for residues in (**A**) the NUDT15 dimer interface and (**B**) the ASPA dimer interface. For each interface residue, the average abundance score (grey) is also shown. If $RMSD_{exposed}$ is smaller than $RMSD_{buried}$, the abundance score substitution profile of the residue resembles the average profile of an exposed residue more than that of a buried residue of the same amino acid type, and vice versa. RMSD values were only calculated for residues with at least five reported abundance scores. To evaluate if $RMSD_{exposed}$ and $RMSD_{buried}$ are considerably different, the substitution profile for every residue was resampled 10,000 times based on the VAMP-seq score averages and standard deviations (see Methods), and $RMSD_{exposed}$ and $RMSD_{buried}$ values were calculated for all of the resampled profiles. For all interface residues analysed here, the standard error of the mean of the resampled RMSD values is too small to be illustrated with error bars. It is also the case that $RMSD_{buried} < RMSD_{exposed}$ or $RMSD_{exposed} < RMSD_{buried}$ for more than 95% of the resampled substitution profiles for all interface residues. (**C**) NUDT15 dimer with one monomer represented by cartoon of backbone and one monomer shown in transparent surface representation. Side chains are shown for all interface residues, and interface residues with $RMSD_{buried}$ smaller than $RMSD_{exposed}$ are shown in orange, while interface residues with $RMSD_{exposed}$ smaller than $RMSD_{buried}$ are shown in green. (**D, E**) ASPA dimer shown from two different angles represented similarly to the NUDT15 dimer.

The online version of this article includes the following figure supplement(s) for figure 5:

**Figure supplement 1.** Discovering residues with abundance score substitution profiles non-typical for their structural environment.

**Figure supplement 2.** Protein structures with solvent-exposed residues found to have a smaller $RMSD_{buried}$ than $RMSD_{exposed}$ shown in dark orange.

1), as described above, allowing us to discover residues for which the structural environment is not necessarily a good indicator of the impact of substitutions on abundance.

Generally, we find many residues to behave as expected, although we identify surface-exposed residues with $\mathrm{RMSD_{buried}} < \mathrm{RMSD_{exposed}}$ and buried residues with $\mathrm{RMSD_{exposed}} < \mathrm{RMSD_{buried}}$ in all proteins. The number of residues for which the abundance score substitution profile does not match the average profile for residues in a similar structural environment depends on the protein. We for instance find 46 out of 351 (13%) exposed residues in PRKN to have $\mathrm{RMSD_{buried}} < \mathrm{RMSD_{exposed}}$ and 15 out of 114 (13%) buried residues in PRKN to have $\mathrm{RMSD_{exposed}} < \mathrm{RMSD_{buried}}$. In the ASPA monomer, we on the other hand find 86 out of 183 (47%) exposed residues with $\mathrm{RMSD_{buried}} < \mathrm{RMSD_{exposed}}$ and 17 out of 130 (13%) buried residues with $\mathrm{RMSD_{exposed}} < \mathrm{RMSD_{buried}}$. The accuracy with which environment indicates mutational tolerance is highest for PRKN and PTEN and lowest for ASPA (*Figure 5—figure supplement 1*). Considering the number of non-interface, surface-exposed residues that we find to have $\mathrm{RMSD_{buried}} < \mathrm{RMSD_{exposed}}$ in NUDT15 and ASPA (*Figure 5—figure supplement 1*, *Figure 5—figure supplement 2*), we cannot exclude that the crystal structure interface residues might be important for maintaining cellular stability in ways that are not just related to the formation of a homodimer.

Given the fact that the six abundance score distributions have considerably different shapes (*Figure 1—figure supplement 1*), some of the identified discrepancies between residue environments and abundance score profiles are perhaps not surprising. For example, since the PTEN abundance score distribution is shifted towards high abundance score values, it makes sense that we identify many residues in PTEN that we structurally classify as buried, but which have a smaller $\mathrm{RMSD_{exposed}}$ than $\mathrm{RMSD_{buried}}$ value. The opposite might be true for solvent-exposed residues in for example ASPA and PRKN that appear to have abundance score substitution profiles most similar to the average profile of buried residues. As discussed above, a number of factors, including both biochemical and technical, likely contribute to producing the different score distributions, and a combination of these factors will likely also give rise to some of the identified mismatches between structural environment and substitution profiles. In spite of this, we have shown that our method is able to identify surface residues for which the mismatches likely are indicators of residues with functions (in this case dimerisation) important for abundance.

In line with this, we focus the rest of our analysis specifically on solvent-exposed residues for which the $\mathrm{RMSD_{buried}}$ is smaller than $\mathrm{RMSD_{exposed}}$, thereby generating hypotheses regarding which exposed residues might be required for maintaining cellular abundance of the proteins. We mapped this group of residues onto the protein structures (*Figure 5—figure supplement 2*) and indeed found that our method works not only for analysis of residues likely involved in homodimerisation, but also for revealing exposed residues that through different types of biochemistry control abundance. In PRKN, solvent-exposed residues with $\mathrm{RMSD_{buried}} < \mathrm{RMSD_{exposed}}$ for example include many of the $Zn^{2+}$-coordinating cysteine and histidine residues that are critical for maintaining abundance (*Clausen et al., 2024*). Our method also identifies that the substitution profile of the solvent-exposed S109 in PRKN is more similar to that of buried than exposed serines, which is in line with the fact that certain mutations of this residue are thought to introduce a solvent-exposed quality control degron in the protein (*Clausen et al., 2024*). Finally, the method appears to be able to capture that certain phosphorylation sites are important for maintaining abundance. Specifically, it has been shown that residues S380, T382, T383, and S385 of PTEN are phosphorylated in the cell and that mutation of the first three residues to alanine increases the cellular degradation rate of the protein (*Vazquez et al., 2000*). With our method, we identify one of these sites, S385, to have $\mathrm{RMSD_{buried}} < \mathrm{RMSD_{exposed}}$ while being exposed in the static structure. The importance of residue S385 has also been discussed in a previous analysis of the VAMP-seq data (*Matreyek et al., 2018*). While the substitution profile of residue T383 does at first sight appear to have $\mathrm{RMSD_{buried}} < \mathrm{RMSD_{exposed}}$, the abundance scores of residue T383 are too noisy to state this with high confidence (see Methods). Since residues S380 and T382 both have less than five reported abundance scores, we did not assess whether our method picks up on these sites as being important for maintaining abundance or not.

Above, we have thus demonstrated that the $\mathrm{RMSD_{buried}}$ and $\mathrm{RMSD_{exposed}}$ values of a residue, which respectively measure the similarity between the substitution profile of the residue and the average substitution profiles of buried and exposed residues, can be used to (*i*) tell apart experimentally relevant from irrelevant structures and (*ii*) reveal residue sites with importance for maintaining cellular

protein stability, thereby generating hypotheses for further studies. Our results highlight the usefulness of the exposure-based substitution matrices beyond abundance score predictions. The approach introduced here is related to other methods that similarly use discrepancies between structure data and single residue substitution mutagenesis data to distinguish correct or relevant structures from decoy or wrongly predicted structures (*Adkar et al., 2012*; *Chiasson et al., 2020*; *Zutz et al., 2021*), and to methods that aim to derive three-dimensional structures using data generated by MAVEs (*Schmiedel and Lehner, 2019*; *Rollins et al., 2019*; *Drake et al., 2024*). Finally, the identified discrepancies between individual and average substitution patterns of solvent-exposed residues point to several aspects of cellular biochemistry that our simple, structure-based abundance framework does not capture, including aspects of post-translational modification and interactions with the cellular degradation machinery after introduction of neo-degrons. The highlighted discrepancies might thus be used to guide future abundance model development, for example via introduction of more complex biochemical input features.

## Conclusions

Several previous studies have combined heterogeneous mutagenesis data from multiple MAVEs to analyse amino acid substitution effects across proteins in a general manner. In this study, we have similarly analysed a combination of six large mutagenesis datasets, however restricting our analysis to a more homogenous set of MAVE data that were all collected with the VAMP-seq method and report on the impact of substitutions on the cellular abundance of proteins. Our results therefore reveal trends in substitution effects for a single molecular phenotype. We have analysed in total 31,614 single residue substitution variant abundance scores in a structure-based fashion through construction of amino acid substitution matrices reporting on the average abundance scores for all possible substitution types.

Importantly, we found that, by using simple structural considerations, it is possible to construct structure context-specific substitution matrices that with relatively high accuracy, at least considering the model simplicity, can predict the impact of residue substitutions on the cellular abundance of proteins. Our substitution matrices can, in fact, rank variant abundance as well as much more complex biophysics- and deep learning-based methods, highlighting the usefulness of simple, interpretable structure features for stability-related variant effect prediction.

As shown in this work and in several previous studies, the cellular abundance of variants correlates with the variant impact on protein folding stability, or $\Delta\Delta G$, explaining in part why simple structural features capture abundance-related variant effects relatively well. We compared abundance score predictions from our substitution matrices to $\Delta\Delta G$ predictions from state-of-the-art stability change models, and while the two prediction types are correlated, we found that they also contain orthogonal information, suggesting that the substitution matrices derived from VAMP-seq data contain abundance-specific signal.

The VAMP-seq-derived substitution matrices can also be used to discriminate between experimentally, in this case cellularly, relevant and irrelevant protein structures. We have specifically shown that two of the six proteins in our combined VAMP-seq dataset, namely NUDT15 and ASPA, likely form homodimers that stabilise the proteins in cells. Using a similar method, the matrices can also point out outlier residues that have unexpected substitution profiles given their structural environment. For example, solvent-exposed residues for which variants increase degradation propensity may be captured. We propose that our substitution matrix-based approach is applicable not only for selecting between monomer and dimer structures as done here, but that it might also be useful in a broader context, for example for selecting between correctly and incorrectly predicted structures.

The average substitution effects described by our substitution matrices generally agree with biochemical expectations. In clustering analyses of the average substitution profiles of different amino acid types, amino acids tend to group together according to physicochemical properties, although with noteworthy exceptions, such as for example the extreme substitutional intolerance of aspartate in buried environments. Moreover, while structural considerations can help rationalise certain variant effects, like those of loop residues with left-handed helix-like backbone conformations, this is not always the case, as revealed by the lack of correlation between secondary structure propensity scales and average abundance scores of residues in $\alpha$-helices.

Given the general consistency between the VAMP-seq-derived substitution matrices and biochemical expectation, we believe that the matrices may generally serve as useful references for how

substitutions will likely affect abundance. While the combined analysis of the six individual VAMP-seq datasets proved useful in this work, future work developing for example prediction methods for variant abundance might benefit from more elaborate considerations regarding and treatment of the experimental and biochemical factors that influence the absolute values of the VAMP-seq abundance scores (*Schulze et al., 2025*).

## Methods
### VAMP-seq data collection and preparation
We obtained VAMP-seq abundance scores and abundance score standard deviations directly from the original VAMP-seq publications (*Matreyek et al., 2018*; *Matreyek et al., 2021*; *Amorosi et al., 2021*; *Suiter et al., 2020*; *Grønbæk-Thygesen et al., 2024*; *Clausen et al., 2024*). In these studies, the abundance scores were calculated in a similar manner so that they should be more directly comparable; first, the abundance of each variant was expressed as a weighted average $W_{var}$ over the frequencies with which the variant was found in each of the four FACS-sorted bins. $W_{var}$ scores for all variants in an experiment were then min-max normalised to values respectively representing the weighted average of nonsense variants and the weighted average of synonymous variants, so that the final abundance score of a variant would be $score_{var} = (W_{var} - W_{nonsense})/(W_{wt} - W_{nonsense})$ (*Matreyek et al., 2018*).

We filtered the six datasets to include only single residue substitution scores in our analysis, that is, we excluded data for synonymous and nonsense variants. We did not perform any additional quality filtering of the data beyond filtering already performed in the original publications, meaning that scores from positions with low mutational depth are included in our analysis. Moreover, since abundance scores were already normalised in the original VAMP-seq work, we used reported scores without further normalisation. PTEN VAMP-seq data has been measured multiple times to increase the mutational completeness of the variant abundance dataset (*Matreyek et al., 2018*; *Matreyek et al., 2021*). In our analysis, we used the most recently published PTEN dataset, which is based on a combination of abundance scores measured in the original and updated PTEN work (*Matreyek et al., 2021*). Finally, we excluded variant abundance scores for the first 28 residues of CYP2C9 from all analyses, since the CYP2C9 N-terminal is a transmembrane helix (*Amorosi et al., 2021*).

### Quantification of dataset noise levels
We quantified the noise levels in the individual datasets using a bootstrapping approach in which we resampled each VAMP-seq dataset 100 times. Each abundance score was resampled from a Gaussian distribution with mean equal to the originally reported abundance score and standard deviation equal to the reported standard deviation of the score. We calculated $r$ (and $r_s$) between the originally reported abundance scores and each resampled set of scores and reported the average of the resulting 100 $r$ (and $r_s$) values in *Figure 1E*. We note that abundance score standard deviations are based on a different number of replicate measurements, both within and across datasets, and that our bootstrapped $r$ values do not in all cases correspond to the correlations between replicate experiments reported in the original VAMP-seq publications, with deviations in particular for TPMT and CYP2C9, where our quantification of the noise levels gives higher numbers for $r$.

### Structure preparation and visualisation
We used combined crystal and AlphaFold2 structures (*Jumper et al., 2021*) in our analysis so that our structural input preserved crystal structure coordinates where possible and at the same time had no missing residues. We created the combined structures by first aligning crystal structures and AlphaFold2 structures using the Matchmaker tool in UCSF ChimeraX (v1.3, *Pettersen et al., 2021*) and then filled out gaps from missing residues in the crystal structure files with coordinates from the AlphaFold2 structures. We then ran the 'alphafold2_ptm' model in ColabFold (v1.5.2, *Mirdita et al., 2022*) with the combined structures as input template structures using UniProt sequences as query sequences (*Supplementary file 1*). We specified to not use a multiple sequence alignment for the structure prediction and otherwise used default settings, including omission of Amber relaxation. We verified that the first ranked ColabFold output structures mostly preserved the crystal structure backbone conformations and then used these structures as inputs for all of our analyses. We took this approach to include as much structural information as possible in our modelling while at the same time making

sure that our input structures resembled the crystal structures in cases where crystal and AlphaFold2 structures deviated with respect to backbone conformation.

The AlphaFold2 structures that we used as input for this approach were predicted with the Alpha-Fold Monomer v2.0 pipeline and downloaded from the AlphaFold Protein Structure Database (*Varadi et al., 2022*), except for in the cases of NUDT15 and ASPA, since these proteins can be found only in their monomeric form in the database. We created the homodimeric assemblies of the proteins with the multimer pipeline 'alphafold2_multimer_v3' in ColabFold using default settings, including template mode 'pdb70' and MSA mode 'mmseqs2_uniref_env'. The top ranked dimer structures generated with this approach were then used as input AlphaFold2 structures for the procedure described above.

We processed structure data files with pdb-tools (v2.4.3, *Rodrigues et al., 2018*). Our structure processing included removal of chains not noted in *Supplementary file 1* and removal of all HETATM entries in the crystal structure files. All our analyses were thus performed without accounting for co-factors or ions bound to the proteins in their crystal structures. We also removed the 28 N-terminal residues of the CYP2C9 structure, as the N-terminal of CYP2C9 is known to form a transmembrane helix (*Amorosi et al., 2021*). In calculations that rely on the monomeric structures of NUDT15 and ASPA, we always used chain A in its conformation in the dimeric structure. All protein structure visualisations were created with ChimeraX.

## Calculation of structural features

We used DSSP (*Kabsch and Sander, 1983*) for secondary structure assignment and calculation of solvent accessible surface area for every residue in the wild-type protein structures. Solvent accessible surface area was normalised to obtain relative solvent accessibility using a theoretically derived maximum accessibility per residue (*Tien et al., 2013*). As in other studies (*Hovmöller et al., 2002*), we used secondary structure assignments from DSSP to classify residues into three secondary structure categories, specifically helix ($\alpha$-, $3_{10}$-, and $\pi$-helix), strand (extended strand and isolated $\beta$-bridge), and loop (hydrogen-bonded turn, bend, and irregular/loop). These definitions were used throughout our work, except in the analysis of abundance score correlations with helix propensity scales (*Figure 4— figure supplement 3*), where we only included data for $\alpha$-helical residues.

We also evaluated contact formation between residues by calculating a WCN for every residue in the wild-type structure environment. Specifically, the WCN for every residue $i$ was calculated as a function of the distances $r_{i,j}$ to all other residues in the protein structure according to the equation

$$\text{WCN}_i = \sum_{j \neq i} s(r_{i,j}) \quad \text{with} \quad s(r) = \frac{1 - \left(\frac{r}{r_0}\right)^6}{1 - \left(\frac{r}{r_0}\right)^{12}}, \tag{1}$$

in which we set $r_0 = 7$ Å. We defined the distance between two residues to be the shortest distance between any pair of heavy atoms in the side chains, except if one residue involved was glycine, in which case we used the shortest distance between any interresidue atom pair. Distances were calculated with the MDTraj (v1.9.7, *McGibbon et al., 2015*) function compute_contacts. We calculated WCN according to *Equation 1*, as we have previously found contact numbers calculated in this manner to correlate with variant abundance (*Cagiada et al., 2023*; *Grønbæk-Thygesen et al., 2024*; *Clausen et al., 2024*; *Gersing et al., 2024*).

## Definitions of buried and exposed residues

We classified residues with an rASA smaller than or equal to 0.1 as structurally buried and residues with an rASA larger than 0.1 as solvent-exposed. In other words, we defined the residue class (RC) as a function of rASA as follows:

$$\text{RC(rASA)} = \begin{cases} \text{buried} & \text{if } \text{rASA} \leq 0.1 \\ \text{exposed} & \text{if } \text{rASA} > 0.1 \end{cases} \tag{2}$$

We assigned a residue class to all residues in the six proteins according to this definition, and to calculate, for example, the 'buried' substitution matrix, we averaged VAMP-seq scores reported for all residues with RC(rASA) = buried. The definition of buried and exposed residues in *Equation 2* is used throughout the manuscript, except in the analysis we performed to choose the 0.1 cutoff value.

The rASA = 0.1 cutoff value was selected using a grid search in which we scanned combinations of rASA and WCN values to maximise the average correlation between experimental abundance scores and abundance scores predicted with the exposure-based substitution matrices. More specifically, we calculated the substitution matrices for buried and exposed residues using a range of rASA and WCN cutoff combinations to classify residues as buried or exposed. rASA cutoff values ($c_{rASA}$) from 0 to 0.5 were scanned together with WCN cutoff values ($c_{WCN}$) from 0 to 20. For this scan, we defined buried residues as residues with a WCN larger than or equal to the WCN cutoff value *and* an rASA smaller than or equal to the rASA cutoff value. Residues not classified as buried were classified as exposed, meaning that exposed residues would have a WCN smaller than the WCN cutoff or an rASA larger than the rASA cutoff. The residue class (RC) definition as a function of both rASA and WCN was thus as follows:

$$
RC(rASA, WCN) = \begin{cases} \text{buried} & \text{if } rASA \leq c_{rASA} \text{ and } WCN \geq c_{WCN} \\ \text{exposed} & \text{if } rASA > c_{rASA} \text{ or } WCN < c_{WCN} \end{cases} \tag{3}
$$

For each combination of rASA and WCN cutoff values, we tested the predictive power of the substitution matrices using leave-one-protein-out cross-validation, that is, we calculated the matrices for each cutoff combination leaving out one of the six proteins and then attempted to predict the abundance scores for the omitted protein. We calculated *r* and the mean absolute error (MAE) between experimental and predicted abundance scores for each protein and then averaged results across proteins. We found that, on average, an rASA cutoff of 0.1 maximises *r* and minimises the MAE when used in combination with WCN cutoff values of 0, 5, and 10, however, with no considerable difference between the three WCN cutoffs (*Figure 2—figure supplement 10A and B*). A WCN cutoff value of 0 effectively corresponds to applying only the rASA cutoff. For simplicity, we therefore used an rASA cutoff of 0.1 (*Equation 2*) and did not consider WCN for exposure-based residue classifications in any other analyses in this work.

## Definition of substitution matrices

To calculate the substitution matrix entry $\bar{s}_{ij}$ for substitutions from amino acid type $i$ to amino acid type $j$, we averaged all individual VAMP-seq scores $s_{ij}$ for this substitution type, such that

$$
\bar{s}_{ij} = \frac{1}{N_{ij}} \sum_{k=1}^{N_{ij}} s_{ij,k}, \tag{4}
$$

where $N_{ij}$ is the total number of VAMP-seq scores for substitutions from residue type $i$ to residue type $j$. To calculate $\bar{s}_{ij}$ for the Global substitution matrix, we simply used all $s_{ij}$ values available in our combined VAMP-seq dataset. Residue environment-specific substitution matrices were calculated using $s_{ij}$ values only for variants of residues sitting in specific environments in the wild-type protein structures. For example, entries in the two substitution matrices that depend on residue burial (*Figure 2*) were calculated according to

$$
\bar{s}_{ij}(rASA) = \begin{cases} \bar{s}_{ij,\text{buried}} & \text{if } rASA \leq 0.1 \\ \bar{s}_{ij,\text{exposed}} & \text{if } rASA > 0.1 \end{cases}, \tag{5}
$$

where $\bar{s}_{ij,\text{buried}}$ and $\bar{s}_{ij,\text{exposed}}$ were calculated with *Equation 4* using VAMP-seq scores $s_{ij}$ for variants of residues that respectively sit in either buried (rASA ≤ 0.1) or exposed (rASA >0.1) environments.

## Calculation of ΔΔG values with Rosetta, RaSP, and ThermoMPNN

We calculated the difference in Gibbs folding free energy between wild-type proteins and all possible single residue substitution variants ($\Delta\Delta G = \Delta G_{\text{variant}} - \Delta G_{\text{wildtype}}$) with Rosetta using the cartesian_ddg

protocol and *opt-nov15* energy function (*Park et al., 2016*). We used an in-house pipeline (https://github.com/KULL-Centre/PRISM/tree/main/software/rosetta_ddG_pipeline, v0.2.1; *Tiemann, 2022*) to first prepare and relax input structures and then calculate energies, which we converted from Rosetta energy units (REU) to kcal/mol with the conversion factor 2.9 REU/(kcal/mol) (*Park et al., 2016*). For evaluation of the effect of a given substitution in dimeric structures, we made the residue substitution in both monomers simultaneously and obtained a ΔΔG for the dimer, which we divided by 2 to get the stability change per residue.

We also calculated variant ΔΔG values using the deep learning-based RaSP (*Blaabjerg et al., 2023*) and ThermoMPNN (*Dieckhaus et al., 2024*) models, both of which estimate single residue substitution ΔΔG values from static protein structures. We used the six wild-type protein structures as inputs to the models, representing NUDT15 and ASPA with their homodimer structures. Since ThermoMPNN was trained to predict monomer stability, we compared the rank correlations between abundance scores and ΔΔG values predicted using either monomer or homodimer structures of NUDT15 and ASPA as input. We found $r_s$ to decrease slightly (smaller is better) from –0.62 to –0.65 when ΔΔG values were predicted with the NUDT15 homodimer rather than the monomer structure. When only variants of NUDT15 interface residues were considered, $r_s$ between ΔΔG values and abundance scores decreased from –0.23 for monomer-based predictions to –0.36 for dimer-based predictions. ΔΔG values calculated with ASPA monomer and homodimer structures correlated equally well ($r_s$ = –0.53) with abundance scores when all variants of ASPA were taken into account. Interface variants of ASPA had $r_s$ values of –0.25 and –0.21 for monomer- and dimer-based evaluations, respectively. Given these results and for consistency with the rest of our analysis, we used the dimer-based ΔΔG predictions for NUDT15 and ASPA from ThermoMPNN throughout our work.

## Methods to compare pairs of substitution profiles

We compared how well ThermoMPNN ΔΔG scores and our exposure-based substitution matrices ranked variant effects at a residue-level in the following way. For every individual residue with VAMP-seq scores reported for at least five variants, we calculated $r_s$ between experimental abundance scores and predicted scores in the form of either ThermoMPNN ΔΔGs or substitution matrix values, thus obtaining $r_{s,\Delta\Delta G}$ and $r_{s,matrix}$, respectively. If $-r_{s,\Delta\Delta G}$ and $r_{s,matrix}$ for a given residue were both positive numbers, we proceeded to resample the experimental residue substitution profile 10,000 times, randomly drawing new abundance scores from variant-specific Gaussian distributions with $\mu$ equal to the experimental VAMP-seq score of a variant and $\sigma$ equal to the VAMP-seq score standard deviation of the same variant, assuming independent errors across variants of a single residue. For every residue, we recalculated $r_{s,\Delta\Delta G}$ and $r_{s,matrix}$ using the 10,000 resampled substitution profiles and then checked how often $-r_{s,\Delta\Delta G} > r_{s,matrix}$ or $-r_{s,\Delta\Delta G} < r_{s,matrix}$. Residues for which at least 95% of resampled substitution profiles satisfied $-r_{s,\Delta\Delta G} > r_{s,matrix}$ were classified as ΔΔG-favoured. On the other hand, we classified residues as matrix-favoured if $-r_{s,\Delta\Delta G} < r_{s,matrix}$ for more than 95% of resampled profiles. For the remaining residues, we report that the two predictors capture variant effects on a residue-level equally well. We only classify residues with at least five variant effect scores and for which $-r_{s,\Delta\Delta G}$ and $r_{s,matrix}$ are both positive, and the sum of residues in the three categories (ΔΔG-favoured, matrix-favoured and 'models similar') is thus smaller than the total number of residues in the six proteins.

We compared the experimental substitution profile of every individual residue to the average profiles of buried and exposed residues in a similar fashion. For every residue, we resampled the residue's substitution profile 10,000 times, drawing new abundance scores from variant-specific Gaussian distributions as described above. We then used the resampled substitution profiles to calculate 10,000 residue-specific $RMSD_{exposed}$ and $RMSD_{buried}$ values. Only when either $RMSD_{exposed} < RMSD_{buried}$ or $RMSD_{exposed} > RMSD_{buried}$ for more than 95% of the resampled substitution profiles did we consider a given residue to have an exposed-like or a buried-like substitution behaviour. We only performed this analysis for residues with substitution profiles consisting of at least five abundance scores. In the text and figures, we only report results for residues that live up to our 95% criterion, unless otherwise noted, such as in the case of residue T383 in PTEN.

## Hierarchical clustering and principal component analysis

Hierarchical clustering of substitution matrix rows and columns was performed and visualised using the seaborn function clustermap. We set function parameters method = 'average' and metric = 'euclidean', meaning that distances between clusters were evaluated by calculating the average distance between all pairs of elements in the clusters using the Euclidean distance as the distance metric. Prior to performing the clustering, we set the average abundance scores for synonymous substitutions equal to one.

We performed PCA of abundance score averages for residues in buried and exposed environments by constructing, for each environment type, a 20x40 matrix that for each of the 20 amino acid types specified the 40 abundance score averages for all substitutions to and from that amino acid residue type. In the reformatted matrices, we set scores for synonymous substitutions equal to one. We then used the scikit-learn class sklearn.decomposition.PCA with default parameters to perform the PCA. The input features were not scaled prior to the analysis.

## Analysis of substitution matrices for helix- and loop-forming residues

We tested the correlation between average abundance score matrices for residues forming $\alpha$-helix using a number of different helix propensity scales (*Pace and Scholtz, 1998*; *Rohl et al., 1996*; *Muñoz and Serrano, 1995*; *Chou and Fasman, 1978*; *Williams et al., 1987*; *Luque et al., 1996*). The scales all report on the residue $\Delta\Delta G$ relative to alanine for helix formation. All numbers for our analysis were taken from a single paper which compared six different helix propensity scales (*Pace and Scholtz, 1998*), and we have here followed the naming of the scales presented in that paper. To test the correlation with our average abundance score matrices, we calculated a helix propensity substitution matrix for each helix propensity scale by subtracting the $\Delta\Delta G$ of the variant residue from the $\Delta\Delta G$ of the reference residue. We then calculated the correlation coefficient between the helix propensity and the average abundance score substitution matrices. For this analysis, we used average abundance score matrices constructed with abundance scores only for residues forming $\alpha$-helix, unlike in all our other analyses in this paper, where our helix matrices include data from residues forming $\alpha$-helix as well as $3_{10}$- and $\pi$-helix.

To analyse the substitution patterns of loop residues adopting backbone conformations similar to those found in left-handed helices, we used the MDTraj functions compute_phi and compute_psi to identify all loop residues for which the backbone $\phi$ and $\psi$ angles simultaneously fell within the ranges $0° < \phi < 180°$ and $-90° < \psi < 90°$ (*Hovmöller et al., 2002*). We excluded residues that were not present in the crystal structures of the proteins from this analysis to make sure that wrongly predicted loop residue conformations from AlphaFold2 did not influence the results.

## Analysis of solvent-exposed residues important for cellular stability

For our analysis of NUDT15 and ASPA homodimerisation, we defined homodimer interface residues as residues that were classified as exposed in the monomer structures and as buried in the dimer structures. To filter out residues which experience minor changes in rASA upon dimerisation, but still change category with the strict 0.1 rASA cutoff used to distinguish between buried and exposed residues, we only included residues in the interface residue category if the change in rASA was more than 0.01 when going from monomer to dimer. The results are not particularly sensitive to the exact cutoff value of 0.01, which was selected based on visual inspection of monomer and dimer structures.

For the analysis of the homodimer interfaces as well as for the analysis of the substitution profiles for all remaining residues in the six proteins, we calculated $\text{RMSD}_{\text{buried}}$ and $\text{RMSD}_{\text{exposed}}$ values only for residue positions for which at least five abundance scores were reported in the experimental datasets. In our results, we thus show no $\text{RMSD}_{\text{buried}}$ and $\text{RMSD}_{\text{exposed}}$ data for any residues with fewer than five abundance scores. We note that all NUDT15 interface residues have at least five abundance scores and that all ASPA interface residues except one (residue 242) have at least five abundance scores.

## Acknowledgements

This work was funded by the Novo Nordisk Foundation challenge program PRISM (Protein Interactions and Stability in Medicine and Genomics, NNF180C0033950, to KL-L). We thank Lene Clausen, Martin Grønbæk-Thygesen, Vasileios Voutsinos, Kristoffer E Johansson, Matteo Cagiada, Amelie Stein, and

Rasmus Hartmann-Petersen and other members of the PRISM centre for productive discussions and suggestions on the topic of protein abundance. We acknowledge access to computational resources at the Biocomputing Core Facility at the Department of Biology, University of Copenhagen.

## Additional information

### Competing interests

Kresten Lindorff-Larsen: Holds stock options in and is a consultant for Peptone Ltd. The other author declares that no competing interests exist.

### Funding

| Funder | Grant reference number | Author |
|---|---|---|
| Novo Nordisk Fonden | NNF18OC0033950 | Kresten Lindorff-Larsen |

The funders had no role in study design, data collection and interpretation, or the decision to submit the work for publication.

### Author contributions

Thea K Schulze, Conceptualization, Data curation, Software, Formal analysis, Investigation, Visualization, Methodology, Writing – original draft; Kresten Lindorff-Larsen, Conceptualization, Resources, Formal analysis, Supervision, Project administration, Writing – review and editing

### Author ORCIDs

Thea K Schulze ⓘD https://orcid.org/0000-0001-6587-6749
Kresten Lindorff-Larsen ⓘD https://orcid.org/0000-0002-4750-6039

Reviewer #1 (Public review): https://doi.org/10.7554/eLife.103721.3.sa1
Reviewer #3 (Public review): https://doi.org/10.7554/eLife.103721.3.sa2
Author response https://doi.org/10.7554/eLife.103721.3.sa3

## Additional files

### Supplementary files

MDAR checklist

Supplementary file 1. Overview of structure and sequence data used for analysis. Protein sequences are given by their UniProt identifiers and crystal structure input files (*Carter et al., 2015*; *Kumar et al., 2015*; *Le Coq et al., 2008*; *Lee et al., 1999*; *Wester et al., 2004*; *Wu et al., 2007*) by their PDB identifiers. We used the PDB file chains and residues noted in the last two columns.

### Data availability

Data and code for reproducing the results presented in this work are available at https://github.com/KULL-Centre/_2024_Schulze_abundance-analysis (copy archived at *Schulze and Lindorff-Larsen, 2026*).

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
