## [Editor Report · eLife Assessment]

This **valuable** study presents a thorough analysis of protein abundance changes caused by amino acid substitutions, using structural context to improve predictive accuracy. By deriving substitution response matrices based on solvent accessibility, the authors demonstrate that simple structural features can predict abundance effects with accuracy comparable to complex methods such as free energy calculations. The strength of the evidence is **convincing**, supported by robust experimental design and comprehensive analyses.

---

## [Referee Report · Reviewer #1 (Public review)]

Significance:

While most MAVEs measure overall function (which is a complex integration of biochemical properties, including stability), VAMP-seq-type measurements more strongly isolate stability effects in a cellular context. This work seeks to create a simple model for predicting the response for a mutation on the "abundance" measurement of VAMP-seq.

Public Review:

Of course, there is always another layer of the onion, VAMP-seq measures contributions from isolated thermodynamic stability, stability conferred by binding partners (small molecule and protein), synthesis/degradation balance (especially important in "degron" motifs), etc. Here the authors' goal is to create simple models that can act as a baseline for two main reasons:

(1) how to tell when adding more information would be helpful for a global model;

(2) how to detect when a residue/mutation has an unusual profile indicative of an unbalanced contribution from one of the factors listed above.

As such, the authors state that this manuscript is not intended to be a state-of-the-art method in variant effect prediction, but rather a direction towards considering static structural information for the VAMP-seq effects. At its core, the method is a fairly traditional asymmetric substitution matrix (I was surprised not to see a comparison to BLOSUM in the manuscript) - and shows that a subdivision by burial makes the model much more predictive. Despite only having 6 datasets, they show predictive power even when the matrices are based on a smaller number. Another success is rationalizing the VAMPseq results on relevant oligomeric states.

Comments on revision:

We have no further comments on this manscript.

---

## [Referee Report · Reviewer #3 (Public review)]

"Effects of residue substitutions on the cellular abundance of proteins" by Schulze and Lindorff-Larsen revisits the classical concept of structure-aware protein substitution matrices through the scope of modern protein structure modelling approaches and comprehensive phenotypic readouts from multiplex assays of variant effects (MAVEs). The authors explore 6 unique protein MAVE datasets based on protein abundance through the lens of protein structural information (residue solvent accessibility, secondary structure type) to derive combinations of context-specific substitution matrices that predict variant impact on protein abundance. They are clear to outline that the aim of the study is not to produce a new best abundance predictor, but to showcase the degree of prediction afforded simply by utilizing structural information.

Both the derived matrices and the underlying 'training' data are comprehensively evaluated. The authors convincingly demonstrate that taking structural solvent accessibility contexts into account leads to more accurate performance than either a structure-unaware matrix, secondary structure-based matrix, or matrices combining both solvent accessibility and secondary structure. The capacity for the approach to produce generalizable matrices is explored through training data combinations, highlighting factors such as the variable quality of the experimental MAVE data and the biochemical differences between the protein targets themselves, which can lead to limitations. Despite this, the authors demonstrate their simple matrix approach is generally on par with dedicated protein stability predictors in abundance effect evaluation, and even outperforms them in a niche of solvent accessible surface mutations, revealing their matrices provide orthogonal abundance-specific signal. More importantly, the authors further develop this concept to creatively show their matrices can be used to identify surface residues that have buried-like substitution profiles, which are shown to correspond to protein interface residues, post-translational modification sites, functional residues or putative degrons.

The paper makes a strong and well-supported main point, demonstrating the widespread utility of the authors' approach, empowered through protein structural information and cutting edge MAVE datasets. This work creatively utilizes a simple concept to produce a highly interpretable tool for protein abundance prediction (and beyond), which is inspiring in the age of impenetrable machine learning models.

---

## [Author Response]

The following is the authors’ response to the original reviews.

**Public Reviews:**

**Reviewer # 1 (Public review):**
Significance:While most MAVEs measure overall function (which is a complex integration of biochemical properties, including stability), VAMP-seqtype measurements more strongly isolate stability effects in a cellular context. This work seeks to create a simple model for predicting the response for a mutation on the "abundance" measurement of VAMPseq.

We thank the reviewer for their evaluation of our work and for their comments and feedback below.

Of course, there is always another layer of the onion, VAMP-seq measures contributions from isolated thermodynamic stability, stability conferred by binding partners (small molecule and protein), synthesis/degradation balance (especially important in "degron" motifs), etc. Here the authors' goal is to create simple models that can act as a baseline for two main reasons:(1) how to tell when adding more information would be helpful for a global model;(2) how to detect when a residue/mutation has an unusual profile indicative of an unbalanced contribution from one of the factors listed above.As such, the authors state that this manuscript is not intended to be a state-of-the-art method in variant effect prediction, but rather a direction towards considering static structural information for the VAMP-seq effects. At its core, the method is a fairly traditional asymmetric substitution matrix (I was surprised not to see a comparison to BLOSUM in the manuscript) - and shows that a subdivision by burial makes the model much more predictive. Despite only having 6 datasets, they show predictive power even when the matrices are based on a smaller number. Another success is rationalizing the VAMPseq results on relevant oligomeric states.

We thank the reviewer for their summary of the main points of our work. Based on the suggestion by the reviewer, we have added a comparison to predictions with BLOSUM62 to our revised manuscript, noting that we have previously compared the BLOSUM62 matrix to a broader and more heterogeneous set of scores generated by MAVEs (Høie et al, 2022).

Specific Feedback:Major points:The authors spend a good amount of space discussing how the six datasets have different distributions in abundance scores. After the development of their model is there more to say about why? Is there something that can be leveraged here to design maximally informative experiments?

We believe that these effects arise from a combination of intrinsic differences between the systems and assay-specific effects. For example, biophysical differences between the systems, such as differences in absolute folding stabilities or melting temperatures, will play a role, as will the fact that some proteins contain multiple domains.

Also, the sequencing-based score for an individual variant in a sort-seq experiment (such as VAMP-seq) depends both on the properties of that variant and on the composition of the entire FACS-sorted cell library. This is because cells are sorted into bins depending on the composition of the entire library, which means that library-to-library composition differences can contribute to the differences between VAMP-seq score distributions.

From our developed models and outliers in predictions from these, it is difficult to tell which of the several possible underlying reasons cause the differences. We have briefly expanded the discussion of these points in the manuscript, and we have moreover elaborated on this in subsequent work (Schulze et al., 2025).

They compare to one more "sophisticated model" - RosettaddG - which should be more correlated with thermodynamic stability than other factors measured by VAMP-seq. However, the direct head-tohead comparison between their matrices and ddG is underdeveloped. How can this be used to dissect cases where thermodynamics are not contributing to specific substitution patterns OR in specific residues/regions that are predicted by one method better than the other? This would naturally dovetail into whether there is orthogonal information between these two that could be leveraged to create better predictions.

We thank the reviewer for this suggestion and indeed had spent substantial effort trying to gain additional biological insights from variants for which MAVE scores or MAVE predictions do not match predicted ∆∆G values. One major caveat in this analysis is that the experimental MAVE scores, MAVE predictions and the predicted ∆∆G values are rather noisy, making it difficult to draw conclusions based on individual variants or even small subsets of variants.

In our revised manuscript, we have added an analysis to discover residue substitution profiles that are predicted most accurately either by a ∆∆G model or by our substitution matrix model, thereby avoiding analysis of individual variant effect scores.

We find that many substitution profiles are predicted equally well by the two model types, but also that there are residues for which one method predicts substitution effects better than the other method. We have added an analysis of the characteristics of the residues and variants for which either the ∆∆G model or the substitution matrix model is most useful to rank variants. Since we only find relatively few residues for which this is the case, we do not expect a model that leverages predicted scores from both methods to perform better than ThermoMPNN across variants.

Perhaps beyond the scope of this baseline method, there is also ThermoMPNN and the work from Gabe Rocklin to consider as other approaches that should be more correlated only with thermodynamics.

We acknowledge that there are other approaches to predict ∆∆G beyond Rosetta including for example ThermoMPNN and our own method called RaSP (Blaabjerg et al, eLIFE, 2023), and we have added comparisons to ThermoMPNN and RaSP in the revised manuscript. We are unsure how one would use the data from Rocklin and colleagues directly, but we note that e.g. RaSP has been benchmarked on this data and other methods have been trained on this data. We originally used Rosetta since the Rosetta model is known to be relatively robust and because it has never seen large databases during training (though we do not think that training of ThermoMPNN and RaSP would be biased towards the VAMP-seq data). We note also that we have previously compared both Rosetta calculations and RaSP with VAMP-seq data for TPMT, PTEN and NUDT15 (Blaabjerg et al, eLIFE, 2023)

I find myself drawn to the hints of a larger idea that outliers to this model can be helpful in identifying specific aspects of proteostasis. The discussion of S109 is great in this respect, but I can't help but feel there is more to be mined from Figure S9 or other analyses of outlier higher than predicted abundance along linear or tertiary motifs.

We agree with these points and have previously spent substantial time trying to make sense of outliers in Figure S9 and Figure S18 (Figure S8 and Figure S18 of revised manuscript). The outlier analysis was challenging, in part due to the relatively high noise levels in both experimental data and predictions, and we did not find any clear signals. Some outliers in e.g. Figure S9 are very likely the result of dataset-specific abundance score distributions, which further complicates the outlier analysis. We now note this in the revised paper and hope others will use the data to gain additional insights on proteostasis-specific effects.

**Reviewer # 2 (Public review):**
Summary:This study analyzes protein abundance data from six VAMP-seq experiments, comprising over 31,000 single amino acid substitutions, to understand how different amino acids contribute to maintaining cellular protein levels. The authors develop substitution matrices that capture the average effect of amino acid changes on protein abundance in different structural contexts (buried vs. exposed residues). Their key finding is that these simple structure-based matrices can predict mutational effects on abundance with accuracy comparable to more complex physics-based stability calculations (ΔΔG).Major strengths:(1) The analysis focuses on a single molecular phenotype (abundance) measured using the same experimental approach (VAMP-seq), avoiding confounding factors present when combining data from different phenotypes (e.g., mixing stability, activity, and fitness data) or different experimental methods.(2) The demonstration that simple structural features (particularly solvent accessibility) can capture a significant portion of mutational effects on abundance.(3) The practical utility of the matrices for analyzing protein interfaces and identifying functionally important surface residues.

We thank the reviewer for the comments above and the detailed assessment of our work.

Major weaknesses:(1) The statistical rigor of the analysis could be improved. For example, when comparing exposed vs. buried classification of interface residues, or when assessing whether differences between prediction methods are significant.

We agree with the reviewer that it is useful to determine if interface residues (or any of the residues in the six proteins) can confidently be classified as buried- or exposed-like in terms of their substitution profiles. Thus, we have expanded our approach to compare individual substitution profiles to the average profiles of buried and exposed residues to now account for the noise in the VAMP-seq data. In our updated approach, we resample the abundance score substitution profile for every residue several thousand times based on the experimental VAMP-seq scores and score standard deviations, and we then compare every resampled profile to the average profiles for buried and exposed residues, thereby obtaining residue-specific distributions of RMSD_buried_ and RMSD_exposed_ values. These RMSD distributions are typically narrow, since many variants in several datasets have small standard deviations. In the revised manuscript, we report a residue to have e.g. a buried-like substitution profile if RMSD_buried_ <RMSD_exposed_ for at least 95% of the resampled profiles. We do not recalculate average scores in substitution matrices for this analysis.

Moreover, to illustrate potential overlap in predictive performance between prediction methods more clearly than in our preprint, we have added confidence intervals in Fig. 2 and Fig. 3 of the revised manuscript. We note that the analysis in Fig. 2 is performed using a leave-one-protein-out approach, which we believe provides the cleanest assessment of how well the different models perform.

(2) The mechanistic connection between stability and abundance is assumed rather than explained or investigated. For instance, destabilizing mutations might decrease abundance through protein quality control, but other mechanisms like degron exposure could also be at play.

We agree that we have not provided much description of the relation between stability and abundance in our original preprint. In the revised manuscript, we provide some more detail as well as references to previous literature explaining the ways in which destabilising mutations can cause degradation. We have moreover performed and added additional analyses of the relationship between thermodynamic stability and abundance through comparisons of stability predictions and predictions performed with our substitution matrix models.

(3) The similar performance of simple matrix-based and complex physics-based predictions calls for deeper analysis. A systematic comparison of where these approaches agree or differ could illuminate the relationship between stability and abundance. For instance, buried sites showing exposed-like behavior might indicate regions of structural plasticity, while the link between destabilization and degradation might involve partial unfolding exposing typically buried residues. The authors have all the necessary data for such analysis but don't fully exploit this opportunity.

This is similar to a point made by reviewer 1, and our answer is similar. We were indeed hoping that our analyses would have revealed clearer differences between effects on thermodynamic protein stability and cellular abundance and have tried to find clear signals. One major caveat in performing the suggested analysis is that both the experimental MAVE scores, ∆∆G predictions and our simple matrix-based predictions are rather noisy, making it difficult to make conclusions based on individual variants or even small subsets of variants.

To address this point, we have added an analysis to discover residue substitution profiles that are predicted most accurately either by a ∆∆G model or by our substitution matrix model, thereby avoiding analysis of individual variant effect scores. We find that many substitution profiles are predicted equally well by the two model types, but we also, in particular, find solvent-exposed residues for which the substitution matrix model is the better predictor. These residues are often aspartate, glutamate and proline, suggesting that surface-level substitutions of these amino acid types often can have effects that are not captured well by a thermodynamical model, either because this model does not describe thermodynamic effects perfectly, or because in-cell effects are necessary to account for to provide an accurate description.

(4) The pooling of data across proteins to construct the matrices needs better justification, given the observed differences in score distributions between proteins (for example, PTEN's distribution is shifted towards high abundance scores while ASPA and PRKN show more binary distributions).

We agree with the reviewer that the differences between the score distributions are important to investigate further and keep in mind when analysing e.g. prediction outliers. However, our results show that the pooling of VAMP-seq scores across proteins does result in substitution matrices that make sense biochemically and can identify outlier residues with proteostatic functions. As we also respond to a related point by reviewer 1, the differences in score distributions likely have complex origins. In that sense, we also hope that our results can inspire experimentalists to design methods to generate data that are more comparable across proteins.

For example, biophysical differences between the systems, such as differences in absolute folding stabilities or melting temperatures will play a role, as will the fact that some proteins contain multiple domains. Also, the sequence-based score for an individual variant in a sort-seq experiment (such as VAMP-seq) depends both on the properties of that variant and from the composition of the entire FACS-sorted cell library. This is because cells are sorted into bins depending on the composition of the entire library, which means that library-to-library composition can contribute to the differences between VAMP-seq score distributions. From our developed models and outliers in predictions from these, it is difficult to tell which of the several possible underlying reasons cause the differences.

Thus, even when experiments on different proteins are performed using the same technique (VAMP-seq), quantifying the same phenomenon (cellular abundance) and done in similar ways (saturation mutagenesis, sort-seq using four FACS bins), there can still be substantial differences in the results across different systems. An interesting side result of our work is to highlight this including how such variation makes it difficult to learn across experiments. We now elaborate on these points in the revised manuscript.

(5) Some key methodological choices require better justification. For example, combining "to" and "from" mutation profiles for PCA despite their different behaviors, or using arbitrary thresholds (like 0.05) for residue classification.

We hope we have explained our methodological choices clearer in the revised paper.

We removed the dependency of the threshold of 0.05 used for residue classification in Fig. S19 of the original manuscript; in the revised manuscript we only report a residue to have e.g. a buried-like substitution profile if RMSD_buried_ <RMSD_exposed_ for at least 95% of the abundance score profiles that we resampled according to VAMP-seq score noise levels, as explained above.

With respect to combining “to” and “from” mutational profiles for PCA, we could have also chosen to analyse these two sets of profiles separately to take potentially different behaviours along the two mutational axes into account. We do not think that there should be anything wrong with concatenating the two sets of profiles in a single analysis, since the analysis on the concatenated profiles simply expresses amino acid similarities and differences in a more general manner.

The authors largely achieve their primary aim of showing that simple structural features can predict abundance changes. However, their secondary goal of using the matrices to identify functionally important residues would benefit from more rigorous statistical validation. While the matrices provide a useful baseline for abundance prediction, the paper could offer deeper biological insights by investigating cases where simple structure-based predictions differ from physics-based stability calculations.This work provides a valuable resource for the protein science community in the form of easily applicable substitution matrices. The finding that such simple features can match more complex calculations is significant for the field. However, the work's impact would be enhanced by a deeper investigation of the mechanistic implications of the observed patterns, particularly in cases where abundance changes appear decoupled from stability effects.

We agree that disentangling stability and other effects on cellular abundance is one of the goals of this work. As discussed above, it has been difficult to find clear cases where amino acid substitutions affect abundance without stability beyond for example the (rare) effects of creating surface exposed degrons. Our new analysis, in which we compare substitution matrix-based predictions to stability predictions, does offer deeper insight into the relationship between the two predictor types and hence possibly between folding stability and abundance.

**Reviewer #3 (Public review):**
"Effects of residue substitutions on the cellular abundance of proteins" by Schulze and Lindorff-Larsen revisits the classical concept of structure-aware protein substitution matrices through the scope of modern protein structure modelling approaches and comprehensive phenotypic readouts from multiplex assays of variant effects (MAVEs). The authors explore 6 unique protein MAVE datasets based on protein abundance (and thus stability) by utilizing structural information, specifically residue solvent accessibility and secondary structure type, to derive combinations of context-specific substitution matrices predicting variant abundance. They are clear to outline that the aim of the study is not to produce a new best abundance predictor but to showcase the degree of prediction afforded simply by utilizing information on residue accessibility. The performance of their matrices is robustly evaluated using a leave-one-out approach, where the abundance effects for a single protein are predicted using the remaining datasets. Using a simple classification of buried and solvent-exposed residues, and substitution matrices derived respectively for each residue group, the authors convincingly demonstrate that taking structural solvent accessibility contexts into account leads to more accurate performance than either a structureunaware matrix, secondary structure-based matrix, or matrices combining both solvent accessibility or secondary structure. Interestingly, it is shown that the performance of the simple buried and exposed residue substitution matrices for predicting protein abundance is on par with Rosetta, an established and specialized protein variant stability predictor. More importantly, the authors finish off the paper by demonstrating the utility of the two matrices to identify surface residues that have buried-like substitution profiles, that are shown to correspond to protein interface residues, posttranslational modification sites, functional residues, or putative degrons.Strengths:The paper makes a strong and well-supported main point, demonstrating the utility of the authors' approach through performance comparisons with alternative substitution matrices and specialized methods alike. The matrices are rigorously evaluated without introducing bias, exploring various combinations of protein datasets. Supplemental analyses are extremely comprehensive and detailed. The applicability of the substitution matrices is explored beyond abundance prediction and could have important implications in the future for identifying functionally relevant sites.

We thank the reviewer for the supportive comments on our work.

Comments:(1) A wider discussion of the possible reasons why matrices for certain proteins seem to correlate better than others would be extremely interesting, touching upon possible points like differences or similarities in local environments, degradation pathways, posttranslation modifications, and regulation. While the initial data structure differences provide a possible explanation, Figure S17A, B correlations show a more complicated picture.

We agree with the reviewer that biochemical and biophysical differences between the proteins might contribute to the fact that some matrices correlate better than others. We also agree that it would be very interesting to understand these differences better. While it might be possible to examine some of the suggested causes of the differences, like differences or similarities in local environments, we have generally found that noise and differences in score distributions make such analyses difficult (see also responses to reviewers 1 and 2). For now, we will defer additional analyses to future work.

(2) The performance analysis in Figure 2D seems to show that for particular proteins "less is more" when it comes to which datasets are best to derive the matrix from (CYP2C9, ASPA, PRKN). Are there any features (direct or proxy), that would allow to group proteins to maximize accuracy? Do the authors think on top of the buried vs exposed paradigm, another grouping dimension at the protein/domain level could improve performance?

We don’t currently know if any protein- or domain-level features could be used to further split residues into useful categories for constructing new substitution matrices, but it is an interesting suggestion. We note that every substitution matrix consists of 380 averages, and creating too many residue groupings will cause some matrix entries to be averaged over very few abundance scores, at least with the current number of scores in the pooled VAMP-seq dataset. For example, while previous work has shown different mutational effects e.g. in helices and sheets (as one would expect), we find that a model with six matrices ({buried,exposed}x{helix,sheet,other}) does not lead to improved predictions (Fig. 2C), presumably because of an unfavourable balance between parameters and data.

(3) While the matrices and Rosetta seem to show similar degrees of correlation, do the methods both fail and succeed on the same variants? Or do they show a degree of orthogonality and could potentially be synergistic?

These are good questions and are related to similar questions from reviewers 1 and 2. In the revised manuscript, we have added additional analyses of differences between predictions from our substitution matrix model and a stability model, and we indeed find that the two methods show a degree of orthogonality. However, since we identify only relatively few residues for which one method performs better than the other, we don’t expect a synergistic model to outperform the stability predictor across all variants in any of the six proteins.

Overall, this work presents a valuable contribution by creatively utilizing a simple concept through cutting-edge datasets, which could be useful in various.
**Reviewing Editor:**
As discussed in more detail below, to strengthen the assessment, the authors are encouraged to:(1) Include more thorough statistical analyses, such as confidence intervals or standard errors, to better validate key claims (e.g., RMSD comparisons).(2) Perform a deeper comparison between substitution response matrices and ΔΔG-based predictions to uncover areas of agreement or orthogonality(3) Clarify the relationship between structural features, stability, and abundance to provide more mechanistic insights.

As discussed above and below, we have added new analyses and clarifications to the revised manuscript.

**Reviewer #1 (Recommendations for the authors):**
Minor points:Why is a continuous version of the contact number used here, instead of a discrete count of neighbouring residues? WCN values of the residues in the core domain can be affected by residues far away (small contribution but not strictly zero; if there are many of them, it adds up).

We have previously found WCN, which quantifies residue contact numbers in a continuous manner, to be a useful input feature for a classifier that determines whether individual residues are important for maintaining protein abundance or function (Cagiada et al, 2023). We have also found WCN and the cellular abundance of single substitution variants to correlate well in individual analyses of different proteins (Grønbæk-Thygesen et al., 2024; Gersing et al., 2024; Clausen et al., 2024).

We have calculated the WCN as well as a contact number based on discrete counts of neighbouring residues for the six proteins in our dataset. When distances between residues are evaluated in the same way (i.e. using the shortest distance between any pair of heavy atoms in the side chains), and when the cutoff value used for the discrete count is equal to the r_0_ of the WCN function, the continuous and discrete evaluations of residue contact numbers are highly and linearly correlated, and their rank correlation with the VAMP-seq data are very similar. We only observe minor contributions from residues far away in the structure on the WCN.

Typos in SI figure captions e.g. Figure S8-11 "All predictions were performed using using...."

Thank you for pointing this out. We have corrected the typos in Figure S8-11 (Figure S7-S10 in the revised manuscript).

Personally, I'd appreciate a definition of these new substitution matrices under the constraints of rASA/WCN values. It was unclear to me until I read the code but we think that the definition is averaging the substitution matrix based on the clusters they are assigned to. If so, this could be straightforwardly defined in the method section with a heaviside step function.

We have added a definition of the “buried” and “exposed” substitution matrices as a function of rASA in the methods section (“Definitions of buried and exposed residues” and “Definition of substitution matrices”) of the manuscript, as well as a definition of how we classified residues as either buried or exposed using both rASA and WCN as input. Our final substitution matrices, as shown in e.g. Fig. 2, do not depend on the WCN; only the substitution matrix results in Figure S6 (Figure S20 in the revised manuscript) depend on both WCN and rASA.

**Reviewer #2 (Recommendations for the authors):**
The following suggestions aim to strengthen the analysis and clarify the presentation of your findings:(1) Specific analyses to consider:(1.1) Analyze buried positions where the exposed matrix performs better. Understanding these cases might reveal properties of protein core regions that show unexpected mutational tolerance.

We agree with the reviewer that a more detailed analysis of buried residues with exposed-like substitution profiles would be very interesting.

We note that for proteins where the VAMP-seq score distribution is shifted towards high values (as it is the case for PTEN, TPMT and CYP2C9), our identification of such residues may be a result of the score distribution differences between the six datasets. To confidently identify mutationally tolerant core regions, it would be best to (a) correct for the distribution differences prior to the analysis or (b) focus the analysis on residues that fall far below the diagonal in Figure S18.

In additional data (which can be found at https://github.com/KULL-Centre/_2024_Schulze_abundance-analysis) ,we provide, for each of the proteins, a list of buried residues for which RMSD_exposed_ <RMSD_buried_ (for more than 95% of resampled substitution profiles, as described under 1.6). We have not analysed these residues further.

(1.2) A systematic comparison of matrix-based vs. ΔΔG-based predictions could help understand both exposed sites that behave as buried (as analyzed in the paper) and buried sites that behave as exposed (1.1), potentially revealing mechanisms underlying abundance changes.

In our revised manuscript, we have added additional analyses to compare matrixbased and ΔΔG-based predictions, focusing on exposed sites for which one prediction method captures variant effects on abundance considerably better the other prediction method. We have not investigated buried sites with exposed-like behaviour any further in this work.

(1.3) Explore different normalization approaches when pooling data across proteins. In particular, consider using log(abundance score): if the experimental error in abundance measurements is multiplicative (which can be checked from the reported standard errors), then log transformation would convert this into a constant additive error, making the analysis more statistically sound.

As we answer below to point 2.2, the abundance scores are, within each dataset, min-max normalised to nonsense and synonymous variant scores, and the score scale is thus in this way consistent across the six datasets. We have explained above and in the revised manuscript that abundance score distribution differences across datasets are likely partially a result of the FACS binning of assay-specific variant libraries. Using only the VAMP-seq scores (that is, without further information about the individual experiments), we cannot correct for the influence of the sorting strategy on the reported scores. A score normalisation across datasets that places all data points on a single scale would require inter-dataset references variant scores, which we do not have. We note that in a subsequent manuscript (Schulze et al, bioRxiv, 2025) we have attempted to take system- and experimentspecific score distributions into account. We now refer to this work in the revised manuscript.

(1.4) Consider using correlation coefficients between predicted and observed abundance profiles as an alternative to RMSD, which is sensitive to the absolute values of the scores.

We agree with the reviewer that using correlation coefficients to compare substitution profiles might also be useful, in particular for datasets with relatively unique VAMP-seq score distributions, such as the ASPA dataset. To explore this idea, we have repeated the analysis presented in Fig. S18 using the Pearson correlation coefficient r rather than the RMSD.

As in Fig. S18, we derive r_buried_ and r_exposed_ for every residue in the six proteins, specifically by calculating r between the abundance score substitution profile of every individual residue and the average abundance score substitution profiles of buried and exposed residues. VAMP-seq data for the protein for which r_buried_ and r_exposed_ are evaluated is omitted from the calculation of average abundance score substitution profiles, and we use only monomer structures to determine whether residues are buried or exposed.

We show the results of this analysis in an Author response image 1 below. In each panel of the figure, r_buried_ and r_exposed_ are shown for individual residues of a single protein. Blue datapoints indicate residues that are solvent-exposed in the wild-type protein structures, and yellow datapoints indicate residues that are buried in the wild-type structures. Residues for which it is not the case that r_buried_ < r_exposed_ or r_exposed_<r_buried_ in more than 95% of 1000 resampled residue substitution profiles (see explanation of resampling method above) are coloured grey. “Acc.” is the balanced classification accuracy, calculated using all non-grey datapoints, indicating how many buried residues have buried-like substitution profiles (r_exposed_<r_buried_) and how many solvent-exposed residues have exposed-like substitution profiles (r_buried_ < r_exposed_). The classification accuracy per protein in this figure cannot be compared to the classification accuracy of the same protein in Fig. S18, since the number of datapoints used in the accuracy calculation differ between the r- and RMSD-based analyses.

Comparing the r-based approach to the RMSD-based approach (Fig. S18), it is clear that the r-based method is less robust than the RMSD-based method for noisy and incomplete datasets. For the noisiest and most mutationally incomplete VAMP-seq datasets (i.e., PTEN, TPMT and CYP2C9) (Fig. 1), there are relatively few residues for which we with high confidence can determine if the substitution profile is more buried- or more exposed-like. When the VAMP-seq data is less noisy and has high mutational completeness, the r-based method becomes more robust and may thus be relevant in potential future work on new VAMP-seq data with small error bars.

In conclusion, we find that RMSD-based approach to compare substitution profiles is more robust than an r-based approach for several of the VAMP-seq datasets that are included in our analysis. We do believe than an approach based on the correlation coefficient, or potentially several metrics, could be relevant to use, since abundance score distributions from VAMP-seq datasets can differ significantly across datasets. So as not to increase the length of the main text of our manuscript, we have not added this analysis to the revised manuscript.

(1.5) Consider treating missing abundance scores as zero values, as they might indicate variants with very low abundance, rather than omitting them from the analysis.

This suggestion would be most relevant for the PTEN, TPMT and CYP2C9 datasets, which all have a relatively small average mutational depth and completeness, as shown in Fig. 1B and 1C. To assess if setting missing abundance scores as zero values would be reasonable, we have compared the distributions of predicted ΔΔG values (from RaSP and ThermoMPNN) and of predicted abundance scores (from our exposure-based substitution matrices) for variants with reported and missing VAMP-seq data. We show the result in Author response image 2, with data aggregated across the six protein systems:

**Author response image 2. sa3fig2:** 

We find that variants with and without VAMP-seq data have similar ΔΔG score distributions and similar predicted abundance score distributions, and there is thus no clear enrichment of predicted loss of abundance for variants with missing VAMP-seq scores. This suggests that missing abundance scores do not necessarily indicate very low abundance. One cause of missing data might instead be problems with library generation (Matreyek et al, 2018, 2021).

We show in Fig. S9 (Fig. S8 of the revised manuscript) that predicted scores for variants with experimental abundance scores of 0 are often overestimated for NUDT15, ASPA and PRKN, but this is not so much a problem for PTEN, TMPT and CYP2C9, the datasets with most missing scores. The lack of an enrichment of low abundance variants from the various predictors would thus still support that missing scores do not necessarily indicate low abundance.

(1.6) Develop a proper statistical framework for comparing buried vs exposed predictions (whether using RMSD or correlations), including confidence intervals, rather than using arbitrary thresholds.

As explained above and in the methods section of our revised manuscript, we have expanded our approach to compare the substitution profile of a residue to the average profiles of buried and exposed residues, and our method now accounts for the noise in the VAMP-seq data, making the analysis more statistically rigorous. In our expanded approach, we compare the substitution profiles of individual residues to the average profiles for buried and exposed residues 10,000 times per residue to get a residue-specific distribution of RMSD_buried_ and RMSD_exposed_ values. Individual RMSD_buried_ and RMSD_exposed_ values are calculated by resampling abundance scores from a Gaussian distribution defined by the experimentally reported abundance score and abundance score standard deviation per variant. We now only report a residue to have e.g. a buried-like substitution profile if RMSD_buried_ < RMSD_exposed_ in at least 95% of our samples. We do not recalculate average scores in substitution matrices for this analysis. We have updated the plots in our manuscript, e.g. in Fig. S18 and S19 of the revised version, to indicate which residues are confidently classified as buried- or exposed-like.

(2) Presentation improvements:(2.1) In Figure 4, consider removing the average abundance scores, which are not directly related to the RMSD comparison being shown.

We have decided to keep the average abundance scores in Fig. 4 (now Fig. 5), as we find the average abundance scores useful for guiding interpretation of the RMSD values. For example, an unusually small average abundance score with a relatively small standard deviation may explain a case where RMSD_buried_ and RMSD_exposed_ are both large. This is for example the case for residue G185 in ASPA.

In our preprint, the error bars on the average abundance scores in Fig. 4 (now Fig. 5) indicated the standard deviation across the abundance scores that were used to calculate the average per position. We have removed these error bars in the revised manuscript, as we realised that these were not necessarily helpful to the reader.

(2.2) I am assuming that abundance scores are defined as the ratio abundance_variant/abundance_wt throughout the analysis, but I don't think this has been explicitly defined. If this is correct, please state it explicitly. In such case, log(abundance_score) would have a simple interpretation as the difference in abundance between variant and wild-type.

Abundance scores are defined throughout the manuscript as sequence-based scores that have been min-max normalised to the abundance of nonsense and synonymous variants, i.e. abundance_score = (abundance_variant abundance_nonsense)/(abundance_wt–abundance_nonsense). We have described the normalisation of scores to wild-type and nonsense variant abundance in lines 164-166 of the original manuscript. We have now added additional information about the normalisation scheme in the methods section. We note that we did not ourselves apply this normalisation to the data; the scores were reported in this manner in the original publications that reported the VAMP-seq experiments for the six proteins.

(2.3) Consider renaming "rASA" to the more commonly used "RSA" for relative solvent accessibility.

We have decided to keep using “rASA” throughout the manuscript.

(2.4) The weighted contact number function used differs from the established WCN measure (Σ1/rij²) introduced by Lin et al. (2008, Proteins). This should be acknowledged and the choice of alternative weighting scheme justified.

As we have also responded to the first minor point of reviewer 1, we have previously found WCN, as it is defined in our manuscript, to be a useful input feature for a classifier that determines whether individual residues are important for maintaining protein abundance or function (Cagiada et al, 2023). We have also previously found this type of WCN to correlate well with variant abundance of individual proteins, as measured with VAMP-seq or protein fragment complementation assays (Grønbæk-Thygesen et al., 2024; Clausen et al., 2024; Gersing et al., 2024). We acknowledge that residue contact numbers or weighted contact numbers could also be expressed in other ways and that alternative contact number definitions would likely also produce values that correlate well with VAMP-seq data. Since the WCN, as defined in our manuscript, already correlates relatively well with abundance scores, we have not explored whether alternative definitions produce better correlations.

(2.5) Replace the phrase "in the above" with specific references to sections or simply "above" where appropriate. Also, consider replacing many instances of "moreover" with simpler alternatives such as "also" or "in addition" to improve readability.

We have changed several sentences according to this suggestion and hope that we have improved the readability of our manuscript.

**Reviewer #3 (Recommendations for the authors):**
(1) It should be explicitly confirmed earlier that complex structures are used for NUDT15 and ASPA when assessing rASA/WCN. Additionally, it would be interesting to see the effect that deriving the matrices using NUDT15 and ASPA monomers would have.

We have commented on the use of NUDT15 and ASPA homodimer structures earlier in the revised manuscript (specifically already in the subsection Abundance scores correlate with the degree of residue solvent-exposure section).

When residues are classified using monomer rather than dimer structures of NUDT15 and ASPA, there is a small effect on the resulting “buried” and “exposed” substitution matrices. Entries in this set of substitution matrices calculated using either monomer or dimer structures typically differ by less than 0.05, and only a single entry differ by more than 0.1. As expected, the “exposed” matrix tend to contain slightly larger numbers when derived from dimer structures than when derived from monomer structures, meaning that when the interface residues are included in the exposed residue category, the average abundance scores of the “exposed” matrix are lowered. For buried residues, the picture is more mixed, although the overall tendency is that the interface residues make the “buried” matrix contain smaller average abundance scores for dimer compared to monomer structures. These results generally support the use of dimer structures for the residue classification.

We here show the differences between the substitution matrices calculated with dimer or monomer structures of NUDT15 and ASPA and using data for all six proteins in our combined VAMP-seq dataset (average_abundance_score_differece = average_abundance_score_dimers – average_abundance_score _monomers):

**Author response image 3. sa3fig3:** 

We have not explored these alternative matrices further.

(2) While the supplemental analyses are rigorous, the abundance of various metrics being presented can be confusing, especially when they seem to differ in their result. For instance, the discussion of Figure S17 (paragraph starting 428) contains mentions of mean differences but then switches to correlations, while both are presented for all panels. The claim "The datasets thus mainly differ due to differences in substitution effects in buried environments. " is well supported by the observed mean differences, but for Pearson's correlations the average panel A ,B values of buried 0.421 vs exposed 0.427 are hardly different. Which of the metrics is more meaningful, and are both needed?

We agree with the reviewer that the claim that “The datasets thus mainly differ due to differences in substitution effects in buried environments” is not well-supported by the r between the substitution matrices, and we have removed this claim from the text.

Since some datasets share VAMP-seq score distribution features, while others do not, the absolute difference between scores or matrices may be relevant to check for some dataset pairs, while the r may be more relevant to check for other dataset pairs. Hence, we have included both metrics in Fig S17 (Fig S11 in the revised manuscript).

(3) Lines 337-340 - does not feel like S7 is the topic, perhaps the authors meant Figure 2A, B? In general, the supplemental figure references are out of order and panel combinations are sometimes confusing.

We have corrected figures references to now be correct and changed the arrangement of supplemental figures so that they now occur in the correct order. We have looked through the panel combinations with clarity in mind, and hope that the current set of main and supplementary figures balances overview and detail.

(4) Line 363 "are also are also".

We have corrected this typo.